# From LLM-Generated Conjectures to Lean Formalizations: Automated Polynomial Inequality Proving via Sum-of-Squares Certificates

Ruobing Zuo[1]   Hanrui Zhao[2]   Gaolei He[1]   Zhengfeng Yang[1]   Jianlin Wang[3]

## Abstract

Automated proving of polynomial inequalities is a fundamental challenge in automated mathematical reasoning, where rich algebraic structure and a rapidly growing certificate search space hinder scalability. Purely symbolic approaches provide strong guarantees but often scale poorly as the number of variables or the degree increases, due to expensive algebraic manipulations and rapidly growing intermediate expressions. In parallel, LLM-guided methods have made notable progress, particularly on competition-style inequalities with a small number of variables. To address the remaining scalability challenges, we propose NSPI, a neuro-symbolic framework that combines the complementary strengths of LLMs and symbolic computation for polynomial-inequality proving. Concretely, an LLM proposes a conjecture in the form of an approximate polynomial Sum-of-Squares (SOS) decomposition; we refine it via symbolic computation to obtain an exact polynomial SOS representation, which directly proves the target inequality, and we further certify the proof in Lean, yielding an end-to-end pipeline from heuristic discovery to machine-checked proof. Experiments on challenging benchmarks involving polynomials with up to 10 variables demonstrate the effectiveness and scalability of the proposed method.

## 1. Introduction

Polynomial inequalities play a fundamental role in areas such as optimization, control, and combinatorics (Kaltofen et al., 2012a; Parrilo, 2000; Maréchal et al., 2015). The automation of complex inequality proving has recently emerged as an important benchmark for evaluating the limits of AI in mathematical reasoning (Trinh et al., 2024; Wei et al., 2024; He et al., 2024; Li et al., 2026). Recent systems such as AlphaGeometry (Trinh et al., 2024; Chervonyi et al., 2025) and Seed-Prover (Chen et al., 2025b;a) have demonstrated impressive performance on Olympiad-level problems, highlighting the potential of large language models (LLMs) for theorem proving. Nevertheless, automated inequality proving remains highly challenging due to the long reasoning chains, vast search spaces, and substantial computational complexity involved, especially for high-dimensional and multivariate cases.

Symbolic computation has long been a cornerstone for proving polynomial inequalities (Yang, 1999; Lasserre, 2002; Uray, 2020; Yang et al., 2023). A widely used approach is based on Sum-of-Squares (SOS) decomposition (Kaltofen et al., 2012a), which transforms the nonnegativity of a polynomial into a semidefinite programming (SDP) (Nie, 2009; Papp & Yildiz, 2019) problem. Modern computer algebra systems further support such pipelines with basic algebraic operations (Heck & Koepf, 1993; De Moura & Bjørner, 2008; Meurer et al., 2017). However, purely symbolic approaches often suffer from poor scalability due to combinatorial explosion and typically fail to produce structured, human-readable proofs. In parallel, recent LLM-based methods have substantially advanced automated formal theorem proving (Lample et al., 2022; Xin et al., 2025; Ren et al., 2025; Lin et al., 2025a; Wang et al., 2025; Lin et al., 2025b) integrated with proof assistants such as Lean (De Moura et al., 2015) and Isabelle (Paulson, 1990). Nevertheless, their performance on complex algebraic inequalities remains limited by the scarcity formalized training data.

To address the above challenges, a promising approach is to integrate neural and symbolic methods, thereby combining the strengths of structured reasoning and symbolic precision while mitigating dependence on large-scale formalized training data (Heule et al., 2016; Trinh et al., 2024; Wei et al., 2024; Li et al., 2025b). However, existing approaches (e.g., AIPS (Wei et al., 2024)) remain limited in two key aspects: they mainly target low-dimensional cases (e.g.,

[1]School of Software Engineering, East China Normal University, Shanghai, China [2]College of Computer Science and Technology, National University of Defense Technology, Changsha, China [3]School of Computer and Information Engineering, Henan University, Kaifeng, China. Correspondence to: Zhengfeng Yang <zfyang@sei.ecnu.edu.cn>.

*Proceedings of the 43rd International Conference on Machine Learning*, Seoul, South Korea. PMLR 306, 2026. Copyright 2026 by the author(s).

ternary or quaternary polynomials), and typically restrict the role of LLMs to search guidance or strategy selection, since directly using LLMs to generate symbolic conjectures remains difficult to control and certify. In this paper, we propose a new neuro-symbolic framework for polynomial inequality proving that targets unconstrained polynomial inequality scenarios, high-dimensional multivariate problems and elevates LLMs to primary conjecture generators. By formulating inequality proving as SOS-based certification and tightly coupling LLM-driven hypothesis generation with symbolic refinement and formal verification, our approach establishes an end-to-end pipeline from heuristic discovery to certified proof, significantly extending the scope of neuro-symbolic automated theorem proving.

The main contributions can be summarized as follows:

- We propose a neuro-symbolic approach for automatically generates complete formal proofs of inequalities through a pipeline of neural conjecture, symbolic correction, and Lean verification.

- We develop a principled reliability bridge that integrates LLM-based heuristic conjecture generation with symbolic exact certification, transforming neural conjectures into machine-checkable proofs and enabling automated inequality proving to scale to higher-dimensional multivariate cases.

- Extensive experiments on 522 challenging inequality problems demonstrate the effectiveness of the proposed method, which outperforms both symbolic computation-based and LLM-assisted approaches, especially on problems with up to 10 variables.

## 2. Related Work

**Symbolic Methods for Inequality Proving.** Polynomial inequality proving has traditionally been approached through symbolic computation. A classical method is based on the SOS methodology (Kaltofen et al., 2012a; Martin-Dorel & Roux, 2017), which reduces nonnegativity certification to the existence of an SOS decomposition and further to solving SDP problem. Recent works (Li et al., 2025a; Pelleriti et al., 2025) have also explored the application of learning-augmented methods to binary classification of SOS and accelerating SDP solvers. Meanwhile, computer algebra systems such as Maple (Heck & Koepf, 1993), Z3 (De Moura & Bjørner, 2008) and SymPy (Meurer et al., 2017) provide fundamental symbolic capabilities, which support algebraic preprocessing and manipulation in inequality-proving pipelines. Nevertheless, these methods typically struggle to produce human-readable reasoning steps, and purely symbolic approaches often suffer from combinatorial explosion as the

problem dimension increases, particularly for multivariate and algebraically intensive polynomial inequalities.

**LLM-based Formal Theorem Proving.** In recent years, LLM-based automated theorem proving has advanced rapidly. Various approaches integrate LLMs with interactive proof assistants such as Lean (De Moura et al., 2015), Coq (Barras et al., 1997) and Isabelle (Paulson, 1990) to produce machine-checkable formal proofs. One line of work fine-tunes models on large-scale corpora of formal proofs to generate proof strategies or local proof tactics (Polu & Sutskever, 2020; Lample et al., 2022; Xin et al., 2025). Another line explores end-to-end generation of complete formal proofs (Ren et al., 2025; Lin et al., 2025a; Wang et al., 2025; Lin et al., 2025b), exemplified by Goedel-Prover (Lin et al., 2025a) ,Kimina Prover (Wang et al., 2025) and DeepSeek-Prover-V2 (Ren et al., 2025). However, LLM-based methods are limited by the scarcity and uneven quality of formal-proof data, and thus remain weak on high-dimensional inequalities.

**Neuro-Symbolic Theorem Proving.** To bridge the scalability limitations of purely symbolic methods and the data bottleneck of purely neural approaches, recent work has explored neuro-symbolic integration for automated theorem proving (Trinh et al., 2024; Wei et al., 2024; Li et al., 2025b; Chervonyi et al., 2025). These methods typically combine neural models with symbolic solvers, using learning-based components to guide or prioritize symbolic reasoning or derivation steps. Representative systems such as Alpha-Geometry (Trinh et al., 2024), AIPS (Wei et al., 2024), and LIPS (Li et al., 2025b) demonstrate the effectiveness of this paradigm in geometry and algebraic inequality proving. In contrast, while existing approaches mainly focus on learning-guided derivation or strategy selection, we treat LLMs as generators of symbolic conjectures and machine-checkable formal verification. Centered on verifiable SOS certificates, our framework establishes an end-to-end neuro-symbolic pipeline for tackling more complex multivariate polynomial inequalities.

## 3. Preliminaries

Let $\mathbb{R}[x] := \mathbb{R}[x_1, \ldots, x_n]$ be the ring of polynomials in $n$ variables with coefficients in the real field $\mathbb{R}$. A polynomial $f(x) \in \mathbb{R}[x]$ is said to be nonnegative or positive semidefinite (PSD) if $f(x) \geq 0$ for all $x \in \mathbb{R}^n$. In this work, we focus on automated proving of unconstrained polynomial nonnegativity: given $f \in \mathbb{R}[\mathbf{x}]$, our goal is to formally prove

$$f(x) \geq 0, \quad \forall x \in \mathbb{R}^n. \tag{1}$$

Beyond establishing (1) mathematically, automated theorem proving additionally requires certificates that are rigorous and machine-checkable, rather than purely numerical or heuristic validations.

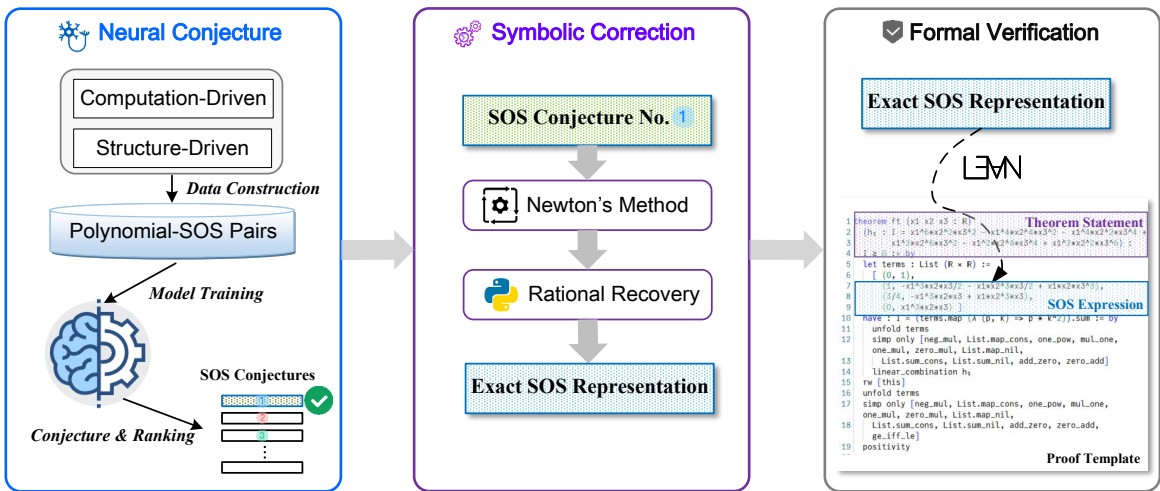

*Figure 1.* **Overview of Neuro-Symbolic SOS-based Polynomial Inequality Proving (NSPI)**. (1) **Neural Conjecture Module**: Non-negative polynomial-SOS representation pairs are constructed using computation-driven and structure-driven approaches. A Large Language Model (LLM) is trained on the constructed data to function as an SOS structure conjecturer, which generates corresponding SOS representations based on the non-negative polynomials and ranks them according to the magnitude of the errors. (2) **Symbolic Correction Module**: An exact SOS representation is derived from the top-ranked SOS structure conjectures through a symbolic computation process that involves Newton iteration and rational recovery. (3) **Formal Verification Module**: Based on the exact SOS representations and predefined Lean proof templates, a complete Lean formal proof is automatically generated.

A widely-used sufficient certificate for (1) is a *sum-of-squares* (SOS) decomposition. A polynomial $f(x)$ is said to be an SOS, if there exist polynomials $f_1(x), \ldots, f_m(x)$ such that

$$f(x) = \sum_{i=1}^{m} f_i(x)^2. \qquad (2)$$

It is immediate $f(x)$ being SOS implies that is nonnegative over $\mathbb{R}^n$, hence an explicit SOS decomposition provides a constructive certificate of (1). Notice that necessarily $f(x)$ must be of even degree $2d$. Let $\mathbf{v}_d(x)$ be the vector

$$\mathbf{v}_d(x) = [1, x_1, x_2, \ldots, x_n, x_1^2, x_1 x_2, \ldots, x_n^d]^\mathsf{T},$$

of all monomials in $x$ and whose degrees are at most $d$, which has dimension $s(d) = \binom{n+d}{d}$. Then $f(x)$ is SOS if and only if there exists a symmetric positive semidefinite matrix $G \succeq 0$ such that

$$f(x) = \mathbf{v}(x)^\mathsf{T} G \mathbf{v}(x). \qquad (3)$$

Equating coefficients in the identity (3) yields a system of linear equations that the entries of $G$ must satisfy. Therefore, determining whether $f(x)$ is SOS can be formulated as a semidefinite feasibility problem:

$$\begin{cases} \text{find} & G \in \mathbb{R}^{s(d) \times s(d)} \\ \text{s.t.} & G \succeq 0, \quad G = G^\mathsf{T}, \\ & f(x) = \mathbf{v}(x)^\mathsf{T} G \mathbf{v}(x). \end{cases} \qquad (4)$$

However, SDP solvers typically return numerical solutions, while formal verification requires exact certificates. This motivates our focus on constructing *exact* SOS certificates that can be directly checked in a proof assistant.

## 4. Methodology

In this section, we introduce **Ne**uro-**Sy**mbolic SOS-based **P**olynomial **I**nequality **P**roving (**NSPI**), a neuro-symbolic framework for automated proving of unconstrained polynomial inequalities. NSPI is designed as a pipeline that integrates LLM-based conjecture with symbolic computation and formal verification. The overall architecture of NSPI is illustrated in Fig. 1, which depicts the following three main stages:

**[Neural Conjecture].** The LLM is employed as a *structure conjecturer* for sum-of-squares (SOS) representations. By combining computation-driven and structure-driven strategies, we construct a diverse dataset of nonnegative polynomials and their corresponding SOS pair forms. The conjecturer is trained on synthetically generated data via a progressive two-stage training scheme, enabling the LLM to more accurately predict plausible SOS structures for given polynomials (see more details in Section 4.1).

**[Symbolic Correction].** Symbolic computation serves as a *precision bridge* in SOS-based proving. At this point, symbolic computation tools are employed to refine the approximate SOS decomposition produced in the previous stage. By combining Newton-type iterative refinement with

rational recovery techniques, numerical solutions are systematically converted into exact rational representations, yielding a precise SOS certificate of the target polynomial. (see more details in Section 4.2).

**[Formal Verification].** This module converts the precise SOS decomposition into machine-checkable Lean proof templates, thereby ensuring that the entire proving process, from conjecture to certificate, is fully verified by a trusted formal kernel. (see more details in Section 4.3).

### 4.1. Neural Conjecture: LLM-Guided SOS Generation

Given an input polynomial, the *neural conjecture* component leverages the structural conjecturing capability of large language models to propose candidate the sum-of-squares decomposition. The overall procedure is organized into two complementary parts as follows:

  i) *Constructing SOS training data*: we construct large-scale polynomial–SOS training pairs by generating SOS polynomials through Gram matrix synthesis, using both computation-driven and structure-driven mechanisms to obtain PSD matrices with controlled coefficient properties. (see Subsection 4.1.1)

  ii) *Training the SOS structure conjecturer*: we train an SOS structure conjecturer via a progressive two-stage scheme, where supervised fine-tuning provides a cold start on the synthetic corpus and curriculum-based reinforcement learning further improves conjecturing performance on harder multivariate instances. (see Subsection 4.1.2)

#### 4.1.1. SOS DATA CONSTRUCTION METHOD

Constructing nonnegative polynomials and and their corresponding SOS representations in a systematic and numerically well-behaved manner is a nontrivial task. A straightforward approach is to randomly sample polynomials $f_i(x)$ and construct $f(x) = \sum_i f_i(x)^2$. However, this naive strategy suffers from two major limitations: (1) the coefficients of the generated $f_i(x)$ are typically non-integer, which makes the resulting polynomial inconsistent with the typical symbolic representations that require integer coefficients, and (2) the expansion of randomly generated squared polynomials often leads to severe coefficient swelling, producing polynomials with unreasonably large or unbalanced coefficients.

To address these issues, we develop a novel data construction method grounded in the algebraic structure of SOS decomposition. As reviewed in Section 3, a polynomial $f(x)$ is SOS if and only if there exists a positive semidefinite (PSD) Gram matrix $\widetilde{G}$ such that $f(x) = \mathbf{v}(x)^\mathsf{T}\widetilde{G}\mathbf{v}(x)$, where $\mathbf{v}(x)$ denotes the vector of monomials. Consequently, the problem of generating suitable polynomial–SOS pairs

reduces to the problem of ***constructing a PSD Gram matrix*** $\widetilde{G}$ with controlled coefficient structure.

Based on this observation, we propose two families of SOS data construction methods, namely *computation-driven* and *structure-driven* approaches. Once a Gram matrix $\widetilde{G}$ is obtained by either approach, the SOS polynomial $f(x)$ is generated by instantiating a monomial basis $\mathbf{v}(x)$.

**(1) Computation-Driven Methods.** We first consider constructing the Gram matrix $\widetilde{G}$ through numerical procedures. The objective is to systematically control the magnitude and precision of the coefficients while ensuring that the resulting $\widetilde{G}$ remains positive semi-definite. The computation-driven methods comprise two main approaches.

**[Explicit Algebraic Construction].** One direct way to obtain a PSD Gram matrix $\widetilde{G}$ is based on spectral shifting. Specifically, we first generate a symmetric integer matrix $G \in \mathbb{S}^m$ and compute its smallest eigenvalue $\lambda_{\min}$. Setting

$$\widetilde{G} = G - kI \succeq 0, \text{ where } k = \lfloor \lambda_{\min} \rfloor \quad (5)$$

yields a matrix $\widetilde{G} \succeq 0$, since its smallest eigenvalue satisfies $\lambda_{\min} - k \geq 0$.

**Alternatively**, a PSD Gram matrix can also be constructed in factored form. Let $L \in \mathbb{Z}^{m \times k}(k \leq m)$ be a sparse integer matrix and let $D \in \mathbb{R}^{k \times k}$ be a positive definite diagonal matrix, where $k$ corresponds to the number of squared terms in the SOS decomposition. Then the matrix

$$\widetilde{G} = L^\mathsf{T}DL \quad (6)$$

is symmetric positive semidefinite by construction.

**[Optimization-Based Approach].** Another computation-driven method constructs the Gram matrix $\widetilde{G}$ through Linear Matrix Inequality (LMI) optimization. This approach employs LMI optimization to compute an approximate SOS representation, followed by controlled adjustment to obtain an exact integer-coefficient solution. Suppose $f(x) \in \mathbb{Z}[x]$ is a randomly selected integer-coefficient polynomial. We first solve the following semidefinite program:

$$\begin{cases} \max \lambda \\ \text{s.t. } f(x) = \mathbf{v}(x)^\mathsf{T}G\mathbf{v}(x) \\ \quad G - \lambda I \succeq 0, \quad G = G^\mathsf{T}. \end{cases} \quad (7)$$

Let $(\lambda, G)$ be an optimal solution. Since $G - \lambda I \succeq 0$, for any integer $k \geq -\lambda$ we have $G + kI \succeq 0$. Choosing $k := \lceil -\lambda \rceil \in \mathbb{Z}$ yields $G + kI \succeq 0$, and therefore

$$\tilde{f}(x) := f(x) + \mathbf{v}(x)^\mathsf{T}(kI)\mathbf{v}(x) = \mathbf{v}(x)^\mathsf{T}(G + kI)\mathbf{v}(x)$$

is SOS polynomial with integer coefficients. Then the Gram matrix is updated as $\widetilde{G} = G + kI$. Appendix B.1 provides the theoretical foundations for the aforementioned computation-driven methods.

*Figure 2.* **Two-stage training of the SOS conjecturer**: (1) **Cold Start**: supervised fine-tuning (SFT) on large-scale synthetic polynomial–SOS pairs; (2) **Progressive reinforcement learning**: curriculum-based GRPO on challenging training data.

**(2) Structure-Driven Methods.** This category constructs the Gram matrix $\widetilde{G}$ by exploiting algebraic matrix structures that guarantee positive semidefiniteness by design, most notably diagonally dominant (dd) matrices and scaled diagonally dominant (sdd) matrices.

Specifically, a matrix $G \in \mathbb{R}^{m \times m}$ is defined as diagonally dominant if it satisfies the following condition: $G_{ii} \geq \sum_{j \neq i} |G_{ij}|, \forall i$. By Gershgorin's circle theorem (Gerschgorin, 1931), any symmetric $G$ that satisfies the diagonal dominance condition is guaranteed to be positive semidefinite. In this case, $\widetilde{G}$ can be directly initialized as a diagonally dominant matrix. Furthermore, we *construct* $\widetilde{G}$ in a structured diagonally dominant form by restricting it to a non-negative combination of $m$ rank-one generators inspired by (Barker & Carlson, 1975):

$$\widetilde{G} = \sum_{i=1}^{m^2} \eta_i U_i, \tag{8}$$

where $\widetilde{G}$ is an $m \times m$ symmetric matrix, $\eta_i \geq 0$, and $U_i = \mathbf{u}_i \mathbf{u}_i^\top$, with $\mathbf{u}_i \in \mathbb{R}^m$ having at most two non-zero components, each equal to $\pm 1$. This representation provides a convenient structural parameterization of PSD Gram matrices, and naturally extends to the scaled diagonally dominant case to introduce greater flexibility. Appendix B.2 provides the theoretical foundations and more details of structure-driven methods.

### 4.1.2. TRAINING THE SOS STRUCTURE CONJECTURER

To obtain an LLM-based SOS structure conjecturer with strong structural reasoning capability, we design a progressive two-stage training scheme. As illustrated in Fig. 2, the procedure consists of two phases. (1) *Cold Start*: **endow** the model with the ability to conjecture SOS structures via supervised fine-tuning (SFT) on the constructed polynomial-SOS pair data. (2) *Progressive Reinforcement Learning*: **further enhance** the model's SOS-structure conjecturing performance via curriculum-based reinforcement learning (RL) on challenging data.

**Cold Start.** We perform SFT on the base model using over one million synthetic data samples. These samples are gen-

erated via the method described in Section 4.1.1, with each instance formulated as a pair $(f(x), S)$, where $f(x)$ denotes a non-negative polynomial and $S$ represents its corresponding SOS decomposition. This stage endows the base model with the foundational capability for SOS structural reasoning, serving as a critical precursor to the subsequent reinforcement learning phase. Further details regarding the synthetic dataset are provided in Appendix D.2.

**Progressive Reinforcement Learning.** We further optimize the model's SOS structural reasoning through progressive reinforcement learning on challenging tasks. Specifically, we curate the training instances that remain unsolved by the cold-started model into an easy-to-hard curriculum and conduct curriculum-based Group Relative Policy Optimization (GRPO) (Shao et al., 2024). The reward function is designed with three critical components:

**[Accuracy Reward].** The accuracy reward encourages SOS structure conjectures with smaller errors compared to the original polynomial. It measures the numerical fidelity of the conjecture and is computed as

$$R_{\text{Accuracy}} = \frac{1}{1 + \alpha \|\mathbf{f}(x) - \hat{\mathbf{f}}(x)\|_2}, \tag{9}$$

where $\mathbf{f}(\mathbf{x})$ represents the original polynomial, $\hat{\mathbf{f}}(\mathbf{x})$ denotes the SOS structure conjecture generated by the model, and $\alpha$ is a scaling factor.

**[Format Reward].** A binary indicator ensuring the output strictly adheres to the predefined SOS structural template and delimiters (e.g., *<SOS Expression>*).

**[Algebraic Structure Penalty].** The algebraic structure consistency penalty is designed to ensure that the set of nonzero monomials in the SOS strucure conjecture closely matches that of the original polynomial, which comprises a *soft penalty* and a *hard penalty*. Appendix D.3 details the specific computation procedure and provides further details on the progressive reinforcement learning procedure.

### 4.2. Symbolic Correction: Exact Rational Recovery

In this part, the *symbolic correction* module serves as a precision bridge between model-generated SOS structure

conjecture and the exact representation required for formal verification, which encompasses two stages: Gauss–Newton refinement and rational recovery. The Gauss-Newton refinement stage is implemented following the methodology proposed in Kaltofen et al. (2012b).

**Gauss–Newton Refinement.** After obtaining the SOS structural conjecture $\hat{f}(x)$ from the *neural conjecture* module, we perform the classical Gauss–Newton refinement to enhance the numerical precision of the SOS representation.

First, the corresponding monomial basis $\mathbf{v}(x)$ is extracted, and an initial floating-point Gram matrix $\mathbf{G}$ is constructed such that $\hat{f}(x) \approx \mathbf{v}(x)^{\mathsf{T}} \mathbf{G} \mathbf{v}(x)$. To refine $\mathbf{G}$ using the Gauss–Newton iteration, we compute the Cholesky decomposition of $\mathbf{G}$:

$$\hat{f}(\mathbf{x}) \approx \mathbf{v}(\mathbf{x})^{\mathsf{T}} L L^{\mathsf{T}} \mathbf{v}(\mathbf{x}) = \sum_{i=1}^{k} \left( \sum_{\alpha} c_{i,\alpha} \mathbf{x}^{\alpha} \right)^2, \quad (10)$$

where $k$ is the rank of the matrix $\mathbf{G}$, and $LL^{\mathsf{T}}$ denotes the Cholesky factorization of the Gram matrix $\mathbf{G}$.

The Gauss–Newton iteration is applied to compute the coefficient correction term $\Delta c_{i,\alpha} \mathbf{x}^{\alpha}$ such that

$$\hat{f}(\mathbf{x}) = \sum_{i=1}^{k} \left( \sum_{\alpha} c_{i,\alpha} \mathbf{x}^{\alpha} + \Delta c_{i,\alpha} \mathbf{x}^{\alpha} \right)^2, \quad (11)$$

where $\Delta c_{i,\alpha} \mathbf{x}^{\alpha}$ represents the perturbation to the polynomial coefficients, and the Gram matrix is updated as $\mathbf{G} + \Delta\mathbf{G}$. The optimization objective is to minimize the backward error:

$$\theta = ||\hat{f}(\mathbf{x}) - \mathbf{v}(x)^{T} \mathbf{G} \mathbf{v}(x)||. \quad (12)$$

Gauss–Newton iteration terminates when $\theta$ falls below a predefined tolerance threshold $\tau$. This process is crucial for the subsequent rational recovery process, ensuring that the obtained SOS representation achieves high precision. Appendix C.1 provides further details of the procedure.

**Rational Recovery.** The numerical Gram matrix $G_N$ obtained from Gauss–Newton refinement contains inherent floating-point errors. Our objective is to transform the matrix into an exact rational PSD matrix that rigorously satisfies the polynomial identity without numerical uncertainty, thereby yielding an exact SOS representation.

According to a classical result on rational recovery (Peyrl & Parrilo, 2008a),[1] we distinguish two recovery regimes

---

[1]If a polynomial admits a Gram matrix lying strictly in the interior of the PSD cone, then there exists a threshold $\delta > 0$ such that any numerical Gram matrix within $\delta$ of the exact solution can be converted into an exact rational PSD matrix by combining rational approximation with orthogonal projection

depending on the numerical rank of the refined solution. (1) *Interior-point case*: when the refined Gram matrix lies strictly in the interior of the PSD cone, we project it orthogonally onto the affine subspace defined by the SOS constraints and then rationalize the resulting matrix. (2) *Boundary case*: when the matrix is numerically rank-deficient, direct matrix rationalization is avoided. Instead, we perform a truncated $LDL^{\top}$ factorization followed by simultaneous Diophantine approximation to recover rational vectors while preserving the rank structure. Appendix C provides further details of Symbolic Correction module.

### 4.3. Formal Verification: Lean Proof Generation

After obtaining the exact SOS certificates, the *formal verification* module generates a complete Lean proof. By integrating pre-defined proof templates with the *neural conjecture* and *symbolic correction* modules, we implement a callable Lean tactic, *llm_ineq*. Given a target polynomial and its exact SOS certificate, *llm_ineq* automatically constructs a complete Lean proof by discharging two obligations:

**Equality between polynomial and SOS certificate.** Lean expands the polynomial and the SOS expression into canonical forms and checks their equality via the *linear_combination* tactic. For example, the code below shows how Lean verifies the equality:

```
have h_eq : p = (terms.map (fun (q, k) =>
    k * q^2)).sum := by
  linear_combination
```

**Nonnegativity of the SOS expansion.** Lean provides several built-in strategies for proving nonnegativity. The nonnegativity of the SOS expression is proved by *Lean*'s *positivity* tactic, which recursively applies standard rules (e.g., *sq_nonneg*, *mul_nonneg*, *add_nonneg*).

```
have h_nn : 0 <= (terms.map (fun (q, k)
    => k * q^2)).sum := by
  positivity
```

Further details on the proof templates and illustrative examples are provided in Appendix E.

## 5. Experiments

### 5.1. A Challenging Benchmark for Multivariate Polynomial Inequalities

Existing mathematical benchmarks for theorem proving **do not** encompass proofs of **high-dimensional multivariate** polynomial inequalities. To fill this gap, we construct **PolyIneqBench**, a challenging benchmark for unconstrained polynomial inequality proving. It contains **522**

*Table 1.* **Comparative Performance on PolyIneqBench (n=3 to 10).** *Pass*: success rate within 1 hour; *t(s)*: mean execution time of solved instances. *DS-Prover-v2* denotes DeepSeek-Prover-v2. Best and second-best results are in **bold** and underlined, respectively.

| Method Name | n=3 | | n=4 | | n=5 | | n=6 | | n=7 | | n=8 | | n=9 | | n=10 | |
|---|---|---|---|---|---|---|---|---|---|---|---|---|---|---|---|---|
| | Pass | t(s) | Pass | t(s) | Pass | t(s) | Pass | t(s) | Pass | t(s) | Pass | t(s) | Pass | t(s) | Pass | t(s) |
| **Symbol-based Method** | | | | | | | | | | | | | | | | |
| Maple | **97.6%** | 24.6 | 39.0% | 107.4 | 26.7% | 451.2 | 8.2% | 54.9 | 6.7% | 821.5 | 6.67% | 456.3 | 3.3% | 3176.6 | 1.7% | 1525.6 |
| Z3 | **97.6%** | **0.6** | 32.5% | 21.8 | 23.3% | 101.9 | 19.7% | 196.4 | 1.7% | 65.0 | 1.7% | 1598.6 | 0.0% | NA | 0.0% | NA |
| **LLM-based Prover** | | | | | | | | | | | | | | | | |
| DS-Prover-v2 | 42.9% | 19.8 | 2.6% | 17.9 | 0.0% | NA | 0.0% | NA | 0.0% | NA | 0.0% | NA | 0.0% | NA | 0.0% | NA |
| Goedel-Prover-v2 | 20.2% | 189.6 | 5.2% | 136.4 | 0.0% | NA | 0.0% | NA | 0.0% | NA | 0.0% | NA | 0.0% | NA | 0.0% | NA |
| Kimina-Prover | 36.9% | 117.0 | 5.2% | 111.7 | 0% | NA | 0.0% | NA | 1.7% | 120.6 | 0.0% | NA | 1.7% | 115.8 | 0.0% | NA |
| **General-purpose LLM** | | | | | | | | | | | | | | | | |
| GPT-5.2 | 26.2% | 56.8 | 10.4% | 86.7 | 1.7% | 51.8 | 4.9% | 154.4 | 3.3% | 75.3 | 3.3% | **55.9** | 3.3% | 68.3 | 1.7% | **49.2** |
| Gemini-3-Pro | 22.6% | 91.2 | 24.7% | 102.4 | **36.7%** | 109.8 | 21.3% | 138.9 | 15.0% | 140.7 | 15.0% | 126.7 | 13.3% | 143.5 | 6.7% | 116.7 |
| DeepSeek-V3.2 | 14.3% | 368.4 | 6.5% | 420.9 | 1.7% | 526.8 | 1.6% | 495.8 | 5.0% | 393.8 | 5.0% | 561.5 | 1.7% | 358.0 | 3.3% | 678.9 |
| **Hybrid system** | | | | | | | | | | | | | | | | |
| LIPS | 71.4% | 82.5 | 27.3% | 131.0 | 13.3% | 141.6 | 11.5% | 195.7 | 8.3% | 150.2 | 8.3% | 153.1 | 1.7% | 148.7 | 0.0% | NA |
| **NSPI (ours)** | 44.1% | 16.3 | **40.3%** | **14.4** | **36.7%** | **20.0** | **29.5%** | **21.4** | **26.7%** | **29.6** | **21.7%** | 63.9 | **15.0%** | **20.8** | **11.7%** | 58.3 |

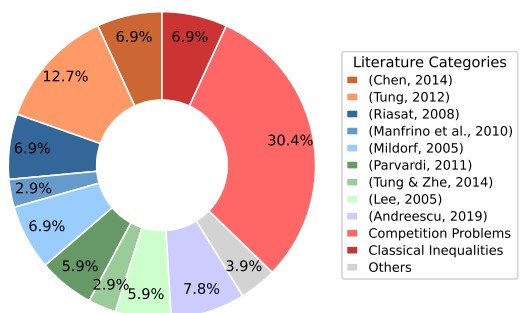

*Figure 3.* Distribution of sample counts across data source categories in PolyIneq-Real.

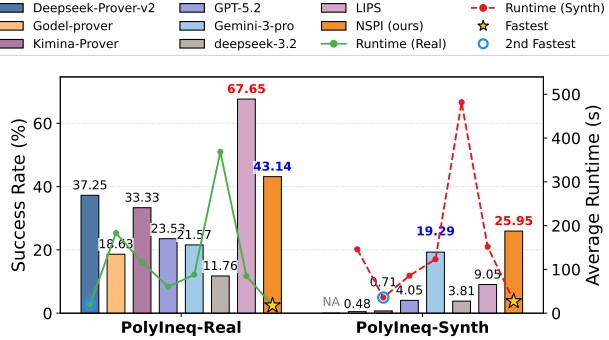

*Figure 4.* Performance of different methods on PolyIneqBench (PolyIneq-Real and PolyIneq-Synth).

complex inequality problems with **3 to 10** variables, all transformed into the form of (1) and formalized in Lean. Among them, 102 problems are collected from international mathematics competitions and other authoritative sources (Chen, 2014; Tung, 2012; Riasat, 2008; Manfrino et al., 2010; Mildorf, 2005; Parvardi, 2011; Tung & Zhe, 2014; Lee, 2005; Andreescu, 2019), forming **PolyIneq-Real**. Fig. 3 shows the distribution of data sources for PolyIneq-Real. Notably, existing competition problems are largely concentrated in the 3 or 4-variable setting. To extend the benchmark to more challenging high-dimensional cases, we synthesize problems with 4 to 10 variables to build **PolyIneq-Synth**. Appendix G.3 provides further details on PolyIneqBench.

### 5.2. Experimental Setup

We compare our method NSPI with several state-of-the-art inequality-proving approaches. Symbolic computa-

tion–based baselines include Maple (Heck & Koepf, 1993) and Z3 (De Moura & Bjørner, 2008). State-of-the-art LLM-based provers include DeepSeek-Prover-V2 (Ren et al., 2025), Goedel-Prover-V2 (Lin et al., 2025a), and Kimina Prover (Wang et al., 2025). We also consider recent general-purpose LLMs, including GPT-5.2, Gemini-3-Pro, and DeepSeek-V3.2 (DeepSeek-AI, 2025), as well as the hybrid system LIPS (Li et al., 2025b). We evaluate all methods on PolyIneqBench using proof success rate and the average runtime of successful proofs, and further examine performance trends as the polynomial degree increases to assess scalability. Appendix G provides additional implementation details and experimental configurations.

### 5.3. Main Results

**Best performance under multivariate scenarios.** Table 1 presents the comparative performance of various baseline

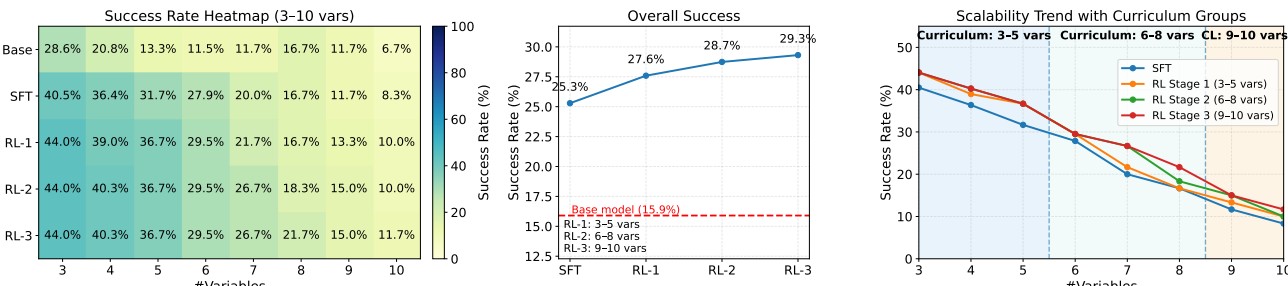

*Figure 5.* **Performance on PolyIneqBench across different training stages**. **Left:** Success rate heatmap showing capability boundary migration. **Middle:** Overall performance gains from base model through RL stages. **Right:** Scalability trends across curriculum groups.

methods on the PolyIneqBench across different numbers of variables $n$. As illustrated, a general downward trend in pass rates is observed for all methods as the problem dimensionality increases. Overall, the proposed NSPI framework outperforms symbol-based methods, pure LLM-based approaches, and other hybrid systems, achieving best performance. Notably, while baseline methods struggle as the variable count scales to $n = 10$, NSPI maintains a consistent pass rate of 11.7%. Our analysis further reveals several key insights: (1) symbolic methods, though proficient in low-dimensional settings, suffer from exponential computational complexity as dimensionality increases and lack the ability to generate human-readable formal proofs; (2) state-of-the-art (SOTA) LLM-based provers encounter significant performance bottlenecks when tackling competition-level polynomial inequalities exceeding 5 variables; and (3) although general-purpose LLMs like Gemini-3-Pro exhibit commendable scalability in certain cases, their overall performance remains inferior to NSPI.

**Challenging synthetic data (PolyIneq-Real vs. PolyIneq-Synth).** We compare the performance of various LLM-assisted methods on PolyIneq-Real and PolyIneq-Synth. Fig. 4 shows a marked decrease in both pass rates and computational efficiency (running time) observed across all methods when tested on synthetic data, validating that the synthesized high-dimensional instances represent more challenging reasoning scenarios. The performance gap on PolyIneq-Real can be partly attributed to the fact that certain polynomials do not admit SOS representations. We evaluated all 102 instances in PolyIneq-Real using an SDP solver (YALMIP), and found that 27 instances do not admit numerical SOS certificates. Moreover, among the instances solvable by LIPS but not by NSPI, 14 problems were also found to have no feasible numerical SOS certificates under YALMIP verification. Notably, on PolyIneq-Real, NSPI outperforms all other LLM-based theorem provers and state-of-the-art general-purpose LLM baselines except LIPS. Under the more rigorous PolyIneq-Synth benchmark, NSPI achieves a **25.95%** pass rate while maintaining the minimum average execution time, thereby underscoring its

superior efficacy and robustness. Notably, NSPI delivers a **2.87-fold** improvement in pass rate over the hybrid LIPS system. In terms of computational throughput, NSPI consistently outperforms its counterparts by maintaining the lowest latency across both benchmarks, yielding a remarkable speedup of up to **10x** compared to several state-of-the-art LLM-based provers.

**Effectiveness of the progressive training process.** Fig. 5 illustrates the performance of the SOS Structure Conjecturer across successive training stages on PolyIneqBench. As illustrated by the color transitions in the left panel of Fig. 5, progressive reinforcement learning (RL) facilitates the migration of the model's **capability boundaries** toward higher-dimensional polynomials. Furthermore, the right panel of Fig. 5 highlights "jump-like" performance improvements at the curriculum boundary points for different data groups. Moreover, training on specific curriculum groups yields cross-group generalization, partially enhancing performance on out-of-distribution instances. It is noteworthy, however, that while the progressive RL process is beneficial, its impact is relatively constrained; as shown in the middle panel of Fig. 5, the large-scale Supervised Fine-Tuning (SFT) during the cold-start phase yields more substantial performance gains for the base model. Overall, large-scale supervised fine-tuning (SFT) in the cold-start phase establishes a strong performance foundation (middle panel), upon which progressive RL provides additional gains by extending generalization to more challenging, higher-dimensional instances.

### 5.4. Ablation Studies

To validate the effectiveness of the individual components of NSPI, we conducted ablation studies under three distinct configurations on PolyIneqBench: (1) removing the synthetic SOS cold start and applying RL directly to the base model; (2) removing progressive RL; and (3) disabling curriculum-based data partitioning during RL. The results in Table 2 show that the cold-start stage trained on large-scale synthetic data is crucial for SOS conjecturing perfor-

mance, and that curriculum-based GRPO further improves the overall results.

*Table 2.* Ablation study of NSPI components. PI-Real and PI-Synth denote PolyIneq-Real and PolyIneq-Synth, respectively.

| SOS Data | GRPO | CL | PI-Real | PI-Synth |
|:---:|:---:|:---:|:---:|:---:|
| ✗ | ✓ | ✓ | 34.31% | 14.76% |
| ✓ | ✗ | ✗ | 40.20% | 21.19% |
| ✓ | ✓ | ✗ | 42.16% | 24.29% |
| ✓ | ✓ | ✓ | **43.14%** | **25.95%** |

For a more comprehensive understanding, Appendix H provides detailed case studies, Appendix I presents additional experimental results, Appendix J provides a systematic analysis of failure modes, and Appendix L provides additional examples where NSPI successfully generates proofs while all baseline methods fail.

## 6. Limitations and Future Work

The proposed NSPI framework is currently applicable only to nonnegative polynomials that have a polynomial Sum-of-Squares (SOS) representation. For polynomials that are nonnegative yet lack an SOS certificate, the framework may currently fail to provide a proof, thereby defining a boundary of its current scope. Furthermore, for polynomials with high degrees or numerous variables, LLM-generated conjectures may suffer from significant coefficient errors, missing terms, or redundancies, thereby limiting the proof success rate.

Several promising directions remain for future investigation. A natural next step is to broaden the class of certificates handled by NSPI. While the current framework focuses on polynomial SOS representations, richer certificates for polynomial nonnegativity exist, including representations as sums of squares of rational functions. Extending NSPI to such broader certificate classes is therefore a theoretically promising direction. In addition, improving the quality of LLM-generated conjectures through richer training data or stronger reasoning scaffolds, such as sparse basis prediction, is another important avenue for future work. More broadly, the proposed "LLM conjecture–symbolic correction–formal verification" pipeline may also extend to related tasks such as exact certification in global polynomial optimization and Lyapunov-based stability verification of polynomial dynamical systems.

## 7. Conclusion

In this paper, we presented NSPI, a neuro-symbolic framework for automated polynomial-inequality proving that combines the complementary strengths of large language models and symbolic computation to provide an end-to-end pipeline from conjecture to certified proof. NSPI leverages an LLM to propose approximate Sum-of-Squares (SOS) decompositions, and refines them via symbolic computation into exact SOS representations that directly prove the target inequalities; we then machine-check these proofs in Lean. Extensive experiments on challenging benchmarks with polynomials of up to 10 variables show that NSPI consistently improves the success rate and efficiency over competitive baseline methods, substantially broadening the practical scope of automated polynomial-inequality proving.

## Acknowledgements

This work was supported in part by the National Key Research and Development Program of China under Grant 2023YFA1009402, the Strategic Priority Research Program of Chinese Academy of Sciences under Grant XDA0480501, and the Natural Science Foundation of Hunan Province under Grant 2026JJ70102.

## Impact Statement

This paper presents work whose goal is to the field of Machine Learning. The development of our approach holds potential for significant impact within the domains of formal verification and automated polynomial inequality proving. While there are various potential societal consequences of this work, none which we feel must be specifically highlighted here.

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

## A. Sum of Squares and Semidefinite Programming

We introduce the relationship between sum-of-squares (SOS) polynomials and semidefinite programming (SDP), which provides the theoretical foundation for the methods employed in this study.

A polynomial $f(\mathbf{x}) \in \mathbb{R}[\mathbf{x}]$ is called a sum-of-squares (SOS) polynomial if it can be expressed as

$$f(\mathbf{x}) = \sum_i f_i(\mathbf{x})^2, \quad \text{where } f_i(\mathbf{x}) \in \mathbb{R}[\mathbf{x}]. \tag{13}$$

*Example* 1. Consider the polynomial $f(\mathbf{x}) = 2x_1^4 + 2x_1^3 x_2 - x_1^2 x_2^2 + 5x_2^4$, $\quad \mathbf{x} = (x_1, x_2) \in \mathbb{R}^2$. Define $f_1(\mathbf{x}) = \frac{1}{\sqrt{2}} \left( 2x_1^2 - 3x_2^2 + x_1 x_2 \right), f_2(\mathbf{x}) = \frac{1}{\sqrt{2}} \left( x_2^2 + 3x_1 x_2 \right)$. Then $f(\mathbf{x})$ admits the SOS decomposition $f(\mathbf{x}) = f_1(\mathbf{x})^2 + f_2(\mathbf{x})^2$, which implies that $f(\mathbf{x}) \geq 0$ for all $\mathbf{x} \in \mathbb{R}^2$.

Semidefinite programming (SDP) is a class of convex optimization problems, whose standard form can be expressed as

$$\begin{aligned}
\text{minimize} \quad & \langle C, G \rangle \\
\text{subject to} \quad & \langle A_i, G \rangle = b_i, \quad i = 1, \ldots, m, \\
& G \succeq 0,
\end{aligned} \tag{14}$$

where $G$ is a symmetric matrix variable and $G \succeq 0$ denotes that $G$ is positive semidefinite. $\langle A, G \rangle = \mathrm{tr}(A^\top G)$ denotes the matrix inner product.

**Theorem A.1.** *(Parrilo, 2000) A multivariate polynomial $f(x)$ in $n$ variables and of degree $2d$ is a sum of squares (SOS) if and only if there exists a symmetric positive semidefinite (PSD) matrix $\widetilde{G}$ such that*

$$f(x) = \mathbf{v}(x)^\top \widetilde{G} \mathbf{v}(x), \tag{15}$$

*where $\mathbf{v}(x) = [1, x_1, x_2, ..., x_n, x_1^2, x_1 x_2, ..., x_n^d]$ is the vector of monomials up to degree d.*

The matrix $\widetilde{G}$ is referred to as a *Gram matrix* of $f(x)$ with respect to the monomial basis $\mathbf{v}(x)$.

*Remark* A.2. The Gram matrix representation is generally not unique, as different choices of the monomial basis or orthogonal transformations of the SOS components yield different PSD matrices $G$.

According to Theorem A.1, expanding the right-hand side of equation Equation (15) and matching the coefficients with those of $f(x)$ yields a system of linear equality constraints on the entries of the Gram matrix $G$. Consequently, the problem of finding an SOS decomposition of the polynomial can be equivalently reformulated as the SDP problem given in Equation (4) of Section 3.

*Example* 2. (Parrilo, 2003) Consider the bivariate quartic polynomial $f(\mathbf{x}) = 2x_1^4 + 2x_1^3 x_2 - x_1^2 x_2^2 + 5x_2^4$, $\quad \mathbf{x} = (x_1, x_2) \in \mathbb{R}^2$. Let the monomial basis vector be $\mathbf{v}(\mathbf{x}) = [x_1^2, \ x_2^2, \ x_1 x_2]^\top$. Then $f(\mathbf{x})$ can be written in Gram form as

$$\begin{aligned}
f(\mathbf{x}) &= \mathbf{v}(\mathbf{x})^\top G \, \mathbf{v}(\mathbf{x}) \\
&= \begin{bmatrix} x_1^2 \\ x_2^2 \\ x_1 x_2 \end{bmatrix}^\top \begin{bmatrix} q_{11} & q_{12} & q_{13} \\ q_{12} & q_{22} & q_{23} \\ q_{13} & q_{23} & q_{33} \end{bmatrix} \begin{bmatrix} x_1^2 \\ x_2^2 \\ x_1 x_2 \end{bmatrix} \\
&= q_{11} x_1^4 + q_{22} x_2^4 + (q_{33} + 2q_{12}) x_1^2 x_2^2 + 2q_{13} x_1^3 x_2 + 2q_{23} x_1 x_2^3.
\end{aligned} \tag{16}$$

Matching coefficients with the target polynomial $f(\mathbf{x})$ yields the following linear equalities:

$$q_{11} = 2, \qquad q_{22} = 5, \qquad q_{33} + 2q_{12} = -1, \qquad 2q_{13} = 2, \qquad 2q_{23} = 0. \tag{17}$$

A positive semidefinite matrix $G \succeq 0$ satisfying (17) can be obtained, e.g., via semidefinite programming. One feasible choice is

$$G = \begin{bmatrix} 2 & -3 & 1 \\ -3 & 5 & 0 \\ 1 & 0 & 5 \end{bmatrix} = L^\top L, \qquad L = \frac{1}{\sqrt{2}} \begin{bmatrix} 2 & -3 & 1 \\ 0 & 1 & 3 \end{bmatrix}.$$

Consequently, $f(\mathbf{x})$ admits the explicit SOS decomposition $f(\mathbf{x}) = \frac{1}{2} \left( 2x_1^2 - 3x_2^2 + x_1 x_2 \right)^2 + \frac{1}{2} \left( x_2^2 + 3x_1 x_2 \right)^2$.

# B. Theoretical Foundations of SOS Data Construction

This part provides the formal theoretical guarantees for the data construction methods described in Section 4.1.1. We focus on the spectral properties and decomposition theorems that ensure the generated Gram matrices $\widetilde{G}$ are positive semidefinite.

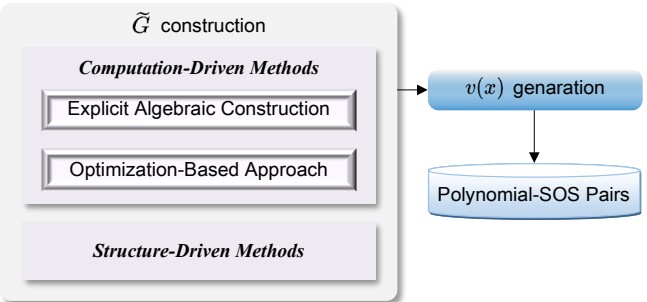

*Figure 6.* **Framework of SOS data construction methods in Section 4.1.1**. The generation of polynomial-SOS pairs is based on the quadratic form $f(x) = \mathbf{v}(x)^\top \widetilde{G} \mathbf{v}(x)$, where the Gram matrix $\widetilde{G}$ is synthesized through either computation-driven or structure-driven approaches.

## B.1. Spectral Foundations for Computation-Driven Methods

The computation-driven methods rely on the relationship between eigenvalues and the PSD property.

**Lemma B.1** (Eigenvalue Shift). *Let $G \in \mathbb{S}^m$ be a symmetric matrix with real entries. For any $k \in \mathbb{R}$, the eigenvalues of $\widetilde{G} = G - kI$ are given by $\lambda_i(\widetilde{G}) = \lambda_i(G) - k$. Consequently, selecting $k \leq \lambda_{\min}(G)$ is a sufficient condition to ensure $\widetilde{G} \succeq 0$.*

To generate Gram matrices with controlled sparsity and coefficient types, we employ a constructive approach based on quadratic forms.

**Lemma B.2.** *Let $L \in \mathbb{Z}^{k \times m}$ be an arbitrary matrix (representing the coefficients of $k$ basis polynomials) and $D = diag(d_1, \ldots, d_k)$ be a diagonal matrix with $d_i > 0$. The resulting matrix*

$$\widetilde{G} = L^\top D L \tag{18}$$

*is guaranteed to be positive semi-definite ($\widetilde{G} \succeq 0$).*

## B.2. Diagonally Dominant (dd) and Scaled Diagonally Dominant (sdd) Cones

Structure-driven methods restrict the search space to specific sub-cones of the PSD cone to simplify formal verification.

**Definition B.3.** (Ahmadi & Majumdar, 2019) A matrix $G \in \mathbb{R}^{m \times m}$ is defined as diagonally dominant (dd) if it satisfies the following condition:

$$G_{ii} \geq \sum_{j \neq i} |G_{ij}|, \quad \forall i. \tag{19}$$

A symmetric matrix $G$ is called scaled diagonally dominant (sdd) if there exists a diagonal matrix $D$ with strictly positive diagonal elements such that the matrix $DGD$ is diagonally dominant.

According to Gershgorin's circle theorem (Gerschgorin, 1931), any symmetric diagonally dominant matrix with non-negative diagonal entries is positive semi-definite.

The following lemma provides the theoretical justification for the extreme ray decomposition used in our framework:

**Lemma B.4.** *(Barker & Carlson, 1975) The cone of $m \times m$ symmetric dd matrices is the set of all matrices that can be represented as:*

$$\widetilde{G} = \sum_{i=1}^{m^2} \eta_i U_i, \quad \eta_i \geq 0 \tag{20}$$

*where each $U_i = \mathbf{u}_i \mathbf{u}_i^\top$ is a rank-one matrix and $\mathbf{u}_i \in \mathbb{R}^m$ is a vector with at most two non-zero components, each belonging to $\{\pm 1\}$, that is, $\mathbf{u}_i \in \{\pm e_k : k \in [m]\} \cup \{\pm e_k \pm e_\ell : 1 \le k < \ell \le m\}$.*

For the construction of scaled diagonally dominant (sdd) matrices, we utilize the properties of diagonal congruence transformation:

**Lemma B.5.** *A matrix $G$ is sdd if there exists a positive definite diagonal matrix $D \succ 0$ such that $DGD$ is dd. By Sylvester's Law of Inertia, diagonal congruence transformations preserve the inertia of a matrix and, in particular, the signs of its eigenvalues. Since $DGD$ is dd, it follows that $DGD \succeq 0$. Consequently, $G = D^{-1}(DGD)D^{-1} \succeq 0$.*

Since $G$ is dd, it admits the decomposition $\sum_{i=1}^{m^2} \eta_i U_i$. Applying the diagonal congruence transformation $\widetilde{G} = D^{-1}GD^{-1}$ yields the sdd matrix $\widetilde{G}$ as a non-negative linear combination of scaled rank-one matrices:

$$\widetilde{G} = \sum_{i=1}^{m^2} \eta_i \widehat{U}_i, \tag{21}$$

where $\widehat{U}_i := D^{-1}U_i D^{-1} = \mathbf{w}_i \mathbf{w}_i^\top$ and $\mathbf{w}_i := D^{-1}\mathbf{u}_i$. Each vector $\mathbf{w}_i$ still has at most two nonzero components, whose magnitudes are determined by the diagonal entries of $D^{-1}$.

## C. Further Details of Symbolic Correction Module

This section presents the theoretical foundations of the symbolic correction module described in Section 4.2. The purpose of this module is to convert the SOS structural conjecture obtained from the SOS conjecture stage into an exact rational sum-of-squares certificate, ensuring suitability for formal verification.

### C.1. Gauss-Newton Refinement for SOS

Suppose a polynomial $f(x) \in \mathbb{R}[x]$ of degree $2d$ is conjectured to admit a sum-of-squares (SOS) decomposition. Let $\mathbf{v}(x)$ denote the monomial basis consisting of all monomials of degree at most $d$. The SOS condition is equivalently expressed in Gram matrix form as $f(x) = \mathbf{v}(x)^\top G\mathbf{v}(x)$, where $G \succeq 0$ is a symmetric positive semidefinite Gram matrix. In practice, numerical solvers only provide an approximate Gram matrix $G$, such that $f(x) \approx \mathbf{v}(x)^\top G\mathbf{v}(x)$.

To reduce numerical errors and improve stability prior to exact rational recovery, we apply a structure-preserving Gauss–Newton refinement.

C.1.1. NUMERICAL NEWTON REFINEMENT OF AN APPROXIMATE GRAM MATRIX

**Factorization-based parameterization.** Assume $G \succeq 0$ numerically and compute a factorization

$$G \approx LL^\top, \tag{22}$$

where $L \in \mathbb{R}^{m \times k}$ and $k = \mathrm{rank}(G)$. Then

$$\mathbf{v}(\mathbf{x})^\top G\,\mathbf{v}(\mathbf{x}) \approx \sum_{i=1}^{k} \ell_i(\mathbf{x})^2, \quad \ell_i(\mathbf{x}) := \sum_\alpha c_{i,\alpha}\mathbf{x}^\alpha. \tag{23}$$

We refine the coefficient vectors $L_{:,i}$ via a Gauss–Newton update so that the induced polynomial $\sum_i \ell_i(\mathbf{x})^2$ matches $f(\mathbf{x})$ as accurately as possible.

**Proposition C.1.** *By applying a Cholesky factorization $G = LL^\top$ or an $LDL^\top$ decomposition to the Gram matrix $G$, and treating the coefficients of the SOS factors as optimization variables, the Gauss–Newton iteration exhibits rapid convergence when applied to SOS problems. Provided that sufficient numerical precision is employed, the backward error $\theta$ can be reduced to arbitrarily small values.*

This yields a high-accuracy numerical approximation, which serves as a reliable foundation for the subsequent rational recovery procedure.

**Residual and stopping criterion.** Let $\mathrm{coeff}(\cdot)$ denote the coefficient vector of a polynomial in the chosen monomial basis. Define the residual

$$r(L) := \mathrm{coeff}\left(\sum_{i=1}^{k} \ell_i(\mathbf{x})^2 - f(\mathbf{x})\right), \qquad \theta := \|r(L)\|_2. \tag{24}$$

Here, $\theta$ denotes the backward error of the numerical Gram matrix $G$. We apply the Gauss–Newton iteration to compute the coefficient correction terms $\Delta c_{i,\alpha}$ so as to minimize $\theta$, as described in Equation (11) of Section 4.2.

Simultaneously, the Gram matrix is updated according to $G \leftarrow G + \Delta G$, where the correction term $\Delta G$ can be expressed as

$$\sum_{i=1}^{k}\left(\sum_{\alpha} \Delta c_{i,\alpha} x^{\alpha}\right)^2 = \mathbf{v}(x)^{\top} \Delta G\, \mathbf{v}(x).$$

The iteration terminates once $\theta < \tau$ for a prescribed tolerance $\tau > 0$.

### C.2. Detailed Formulation of Exact Rational Recovery

The following proposition establishes the fundamental equivalence between the existence of a rational sum-of-squares (SOS) decomposition and the existence of a Gram matrix with rational entries.

**Proposition C.2.** *(Peyrl & Parrilo, 2008a) Let $f(\mathbf{x}) \in \mathbb{Q}[\mathbf{x}]$ be a polynomial, and let $\mathbf{v}(\mathbf{x})$ denote a fixed vector of monomials. Then the following statements are equivalent:*

1. *The polynomial $f(\mathbf{x})$ admits a rational SOS decomposition, that is, there exist polynomials $f_i(\mathbf{x}) \in \mathbb{Q}[\mathbf{x}]$ such that*

$$f(\mathbf{x}) = \sum_{i=1}^{r} f_i(\mathbf{x})^2.$$

2. *There exists a symmetric positive semidefinite Gram matrix $G \in \mathbb{S}^m \cap \mathbb{Q}^{m \times m}$ such that*

$$f(\mathbf{x}) = \mathbf{v}(\mathbf{x})^{\top} G\, \mathbf{v}(\mathbf{x}), \qquad G \succeq 0.$$

For the numerical solution $G_N$ obtained after the Gauss–Newton refinement, when the backward error $\theta$ is sufficiently small, one can recover from $G_N$ an exact rational PSD matrix $\widetilde{G}$, which satisfies the polynomial identity exactly, thereby yielding a certified rational SOS certificate. Which satisfies the following identity:

$$f(x) - \mathbf{v}(x)^{\top} \widetilde{G} \mathbf{v}(x) = 0, \qquad \widetilde{G} \succeq 0. \tag{25}$$

The matrix $G$ is projected onto the following affine hyperplane:

$$\mathcal{X} = \left\{ G \mid G^{\top} = G, f(x) - \mathbf{v}(x)^{\top} G \mathbf{v}(x) = 0 \right\}. \tag{26}$$

Suppose that the affine hyperplane defined by Equation (26) can be represented by a linear system $Ay = b$, where $y$ consist of the entries of $G$. If the matrix $A$ has full row rank, then such a hyperplane is guaranteed to exist.

The recovery strategy depends on whether the refined Gram matrix lies in the interior of the PSD cone.

An exact solution satisfying Equation (25) can be obtained via the following two approaches:

- Case 1: If the matrix $G_N$ is of full rank, the solution is recovered by applying an orthogonal projection method.

- Case 2: Otherwise, a rational vector recovery method is employed.

**Case 1: Interior Point Solution ($G_N$ is full rank).** The solution of Equation (25) lies in the intersection of the affine hyperplane $\mathcal{X}$ and the positive semidefinite cone. To compute $\widetilde{G}$, the corresponding orthogonal projection can be obtained by solving the following least-squares problem.

$$\min_{G \succeq 0} \|G_N - G\|_F^2 \quad \text{s.t. } f(x) = \mathbf{v}(x)^\top G \mathbf{v}(x). \tag{27}$$

Next, to verify whether the recovered rational solution $\widetilde{G}$ is a symmetric positive semidefinite matrix, we compute the exact $LDL^\top$ decomposition of $\widetilde{G}$.

$$f(x) = \mathbf{v}(x)^\top \widetilde{G} \mathbf{v}(x) = \mathbf{v}(x)^\top LDL^\top \mathbf{v}(x) \tag{28}$$

**Theorem C.3.** *Let $G_N$ be the refined numerical solution whose backward error satisfies $\theta < \tau$, and let $Ay = b$ be the linear system associated with the affine hyperplane defined by Equation (26). Suppose that $\widetilde{G}$ is the optimal rational solution of the least-squares Equation (27), and that the matrix $A$ has full row rank. If the minimal eigenvalue $\lambda$ of $\widetilde{G}$ satisfies*

$$\lambda > \|G_N\|_F^2 \kappa_2^2(A) \tau^2, \tag{29}$$

*then $\widetilde{G}$ is an exact solution of Equation (25).*

*Proof.* Since $\widetilde{G}$ is a solution of Problem (27), it clearly satisfies the polynomial identity $f(x) = \mathbf{v}(x)^\top \widetilde{G} \mathbf{v}(x)$. Let $y_N$ and $\widetilde{y}$ denote the vectors consisting of the entries of $G_N$ and $\widetilde{G}$, respectively. From (26), and under the assumption that the matrix $A$ has full row rank, we have

$$\|Ay_N - b\|_2^2 = \theta < \tau, \qquad A\widetilde{y} = b. \tag{30}$$

According to the perturbation result (Golub & Van Loan, 2013) for full-row-rank underdetermined linear systems, the following estimate holds:

$$\|y_N - \widetilde{y}\|_2 \leq (\kappa_2(A)\tau)\|y_N\|_2 + O(\tau^2). \tag{31}$$

Under the assumption that $\lambda > \|G_N\|_F^2 \kappa_2^2(A)\tau^2$, it follows that

$$\|G_N - \widetilde{G}\|_F^2 \leq \|y_N - \widetilde{y}\|_2^2 < \lambda \tag{32}$$

where the inequality $\|y_N - \widetilde{y}\|_2^2 < \lambda$ holds because the higher-order term $O(\tau^2)$ is negligible when $\tau$ is sufficiently small. Let $\widetilde{\lambda}$ denote an eigenvalue of $\widetilde{G}$. By the Wielandt–Hoffman theorem (Golub & Van Loan, 2013), we obtain

$$|\widetilde{\lambda} - \lambda| \leq \|G_N - \widetilde{G}\|_F^2 \leq \lambda, \tag{33}$$

which implies that all eigenvalues of $\widetilde{G}$ are nonnegative. Therefore, we conclude that $\widetilde{G} \succeq 0$.

**Case 2: Boundary Solution ($G_N$ is rank-deficient).** When the numerical Gram matrix $G_N$ obtained via Gauss–Newton refinement is rank-deficient or ill-conditioned (near-singular), the affine hyperplane $\mathcal{X}$ defined by the linear constraints is typically tangent to the boundary of the positive semidefinite (PSD) cone. In such instances, $G_N$ does not reside within the interior of the PSD cone; consequently, direct orthogonal projection may yield a rational matrix $\widetilde{G}$ that violates the semi-definiteness requirement. We provides a rigorous discussion on the structural origins of such cases and presents a theoretical framework for exact recovery based on rational vector reconstruction.

- **Redundant Monomials**: In the construction of the SOS decomposition, a monomial basis that is not strictly necessary may be used. This situation can be avoided by exploring the sparse structure of the polynomial or by removing entire rows and columns of the Gram matrix $G_N$ that correspond to numerically small entries, i.e., eliminating the monomials that should not appear in the polynomial's SOS representation.

- **Intrinsic Singularity**: If the polynomial $f(x)$ attains its global minimum at a nonzero real point $(\xi_1, \ldots, \xi_n)$, then the monomial vector $\mathbf{v}(\xi)$ at this point is a zero vector in the Gram matrix $G_N$, implying that $G_N$ is singular. When the global minimum is attained only at finitely many nonzero points, and the backward error $\theta$ is sufficiently small, the Gram matrix can be rendered full rank by performing Gauss–Newton refinement, after which an orthogonal projection can be applied. In contrast, when the global minimum is attained on some manifolds, we apply the Gauss–Newton iteration to a truncated triangular decomposition of $G_N$. Once the residual is sufficiently small, a rational recovery of $G_N$ is carried out via a simultaneous Diophantine approximation algorithm (Lagarias, 1985).

# D. Neural Conjecture Module Configuration

## D.1. Prompt Template

Here we present the prompt template employed by the SOS structure conjecture module in the *neural conjecture* module. For a given nonnegative polynomial, the SOS structure conjecturer generates a candidate SOS structural conjecture based on an expanded-form representation, with the objective that the proposed SOS structure matches the target polynomial as closely as possible.

---

**Prompt Template for SOS Structure Conjecturer**

**Task:**
You are given a polynomial that is the expanded form of a sum-of-squares (SOS) expression. Your task is to reconstruct a plausible SOS representation whose expanded form matches the given polynomial as closely as possible.

**Instructions:**
1. Analyze the input polynomial carefully, focusing on the coefficients and the combinations of variables involved.
2. Infer possible linear or polynomial terms inside each square in the sum-of-squares expression.
3. Construct a sum of square terms without expanding the squares. Keep the output compact and well-structured.
4. Aim for the expanded form of your SOS expression to closely approximate the coefficients in the original polynomial. Minor numerical deviations are acceptable.
5. If multiple valid SOS decompositions exist, prefer one that is simple, symmetric, and easy to interpret.
6. Include variables and constants inside parentheses when appropriate to match constant terms in the input polynomial.

**Output format:**
Please provide your response in the following structure:
`<SOS Expression>:  <sum_of_squares_expression>`
Where `<sum_of_squares_expression>` is a sum-of-squares term, expressed compactly.
For example:
`(x1 + 1)^2 + (2*x1 + 3*x2)^2`

**Key considerations:**
1. Do not simply rewrite the input polynomial or output expanded terms. The output must be a sum-of-squares expression.
2. Constants inside square terms can be fractional or decimal, as needed, to best approximate the original polynomial.
3. Avoid expanding the squares in the output; always keep terms inside parentheses squared.
4. Aim for clear and interpretable variable groupings. Symmetry and simplicity are preferred if multiple answers fit.
5. Your SOS expression should fully explain the input polynomial's structure and coefficients to the best achievable extent.

**Example:**
Input polynomial:
`x1^2 + 2*x1 + 1`
Output:
`(SOS Expression):  (x1 + 1)^2`
Explanation:
The polynomial is a perfect square trinomial; the SOS reconstruction is exact.

Input polynomial:
`5*x1^2 + 12*x1*x2 + 6*x1 + 9*x2^2 + 9`
Output:
`(SOS Expression):  (x1 + 2.99)^2 + (2*x1 + 3*x2)^2`

Now, please provide the SOS reconstruction for the following polynomial:
**Original polynomial: {polynomial}**

---

As illustrated by the prompt template above, it consists of a task description, multiple explicit instructions specifying the

required output format and content constraints for the large language model, and several illustrative examples provided to guide the generation process.

### D.2. Synthetic Dataset Statistics

During the data construction stage of the *neural conjecture* module, we generate more than 100w synthetic data instances using the four data generation strategies proposed in Section 4.1.1, including both **computation-driven** and **structure-driven** approaches. These synthetic datasets, covering problems with varying numbers of variables ranging from 3 to 10, are used as the training corpus for the neural conjecturer. The training set is disjoint from all evaluation benchmarks.

Notably, we performed a **manual quality inspection** of the synthesized polynomial–SOS pair data via random sampling. For each variable setting from 3 to 10, we randomly sampled 50 instances for review. The inspection results indicate that the synthesized data contain no formatting errors and **encompass polynomial SOS decomposition cases of varying difficulty**, including a substantial proportion of **highly non-trivial instances** that pose significant challenges to conventional symbolic solvers. Additionally, we computed the degree and number of SOS terms of the synthesized data. The statistical results are reported in Fig. 7 and Fig. 8.

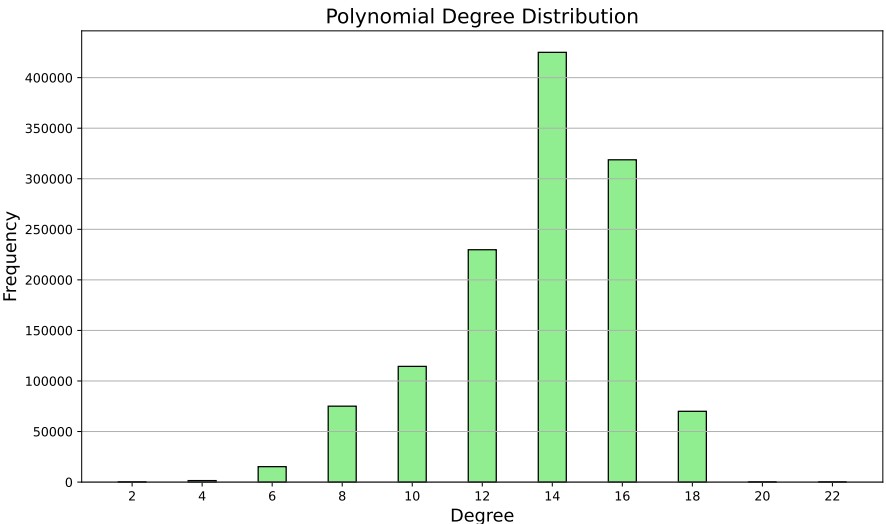

*Figure 7.* Statistical distribution of degrees for synthetic polynomial data.

As illustrated in Fig. 7 and Fig. 8, the synthetic dataset exhibits significant diversity in both algebraic complexity and representation structure. Specifically, the degrees of the synthesized polynomials span a wide range from 2 to 22, ensuring the model is exposed to problems of varying algebraic depths; this reflects that the synthetic data encompasses both relatively simple low-degree polynomials and structurally sophisticated high-degree instances. Furthermore, the distribution of SOS term counts covers a spectrum ranging from basic single-term squares to complex decomposition instances involving up to 15 terms. Such structural diversity indicates that our construction strategies effectively cover a broad difficulty spectrum, preventing the model from over-fitting to specific sparse or low-degree patterns.

### D.3. Additional Details of Progressive Reinforcement Learning Process

For the progressive two-stage training procedure of the SOS-structure conjecturer described in Section 4.1.2, in the second stage we employ a **curriculum-style GRPO** reinforcement learning process. Here, we provide the theoretical foundations and several implementation details.

**Group Relative Policy Optimization (GRPO).** Group Relative Policy Optimization (GRPO) (Shao et al., 2024) is a reinforcement learning algorithm used for fine-tuning large language models (LLMs). Unlike the popular Proximal Policy Optimization (PPO), GRPO enhances efficiency by eliminating the need for a separate value function. It estimates the advantage by normalizing the rewards of a set of responses to the same prompt. Specifically, for each question $q$ in a given set $Q$, a set of responses $\{o_1, \ldots, o_G\}$ is sampled from the old policy $\pi_{\text{old}}$. The reward model then evaluates these responses,

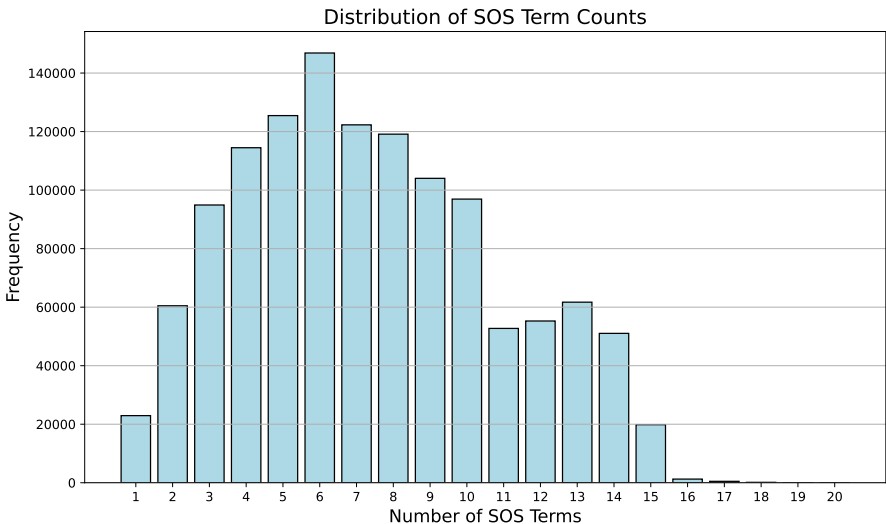

*Figure 8.* Statistical distribution of SOS term counts in synthetic polynomials.

generating rewards $\{r_1, \ldots, r_G\}$, and the advantage is computed as follows:

$$\hat{A}i = \frac{r_i - \text{mean}(r_1, \ldots, r_G)}{\text{std}(r_1, \ldots, r_G)} \tag{34}$$

The policy model $\pi_\theta$ is then optimized by maximizing the following objective:

$$J_{\text{GRPO}}(\theta) = \mathbb{E}_{q \sim Q, \{o_i\}i=1^G \sim \pi_{\text{old}}(O|q)} \frac{1}{G} \sum_{i=1}^{G} \frac{1}{|o_i|} \sum_{t=1}^{|o_i|} \{\min\left[\gamma_{i,t}(\theta)\hat{A}_{i,t}, \text{clip}\left(\gamma_{i,t}(\theta), 1-\epsilon, 1+\epsilon\right)\hat{A}_{i,t}\right] - \beta D_{\text{KL}}(\pi_\theta||\pi_{\text{ref}})\} \tag{35}$$

where $\gamma_{i,t}(\theta) = \frac{\pi_\theta(o_{i,t}|q,o_{i,<t})}{\pi_{\text{old}}(o_{i,t}|q,o_{i,<t})}$ is the importance sampling ratio, $\pi_{\text{ref}}$ represents the reference model, $\pi_{\text{old}}$ is the policy used to sample the responses, and $D_{\text{KL}}(\pi_\theta||\pi_{\text{ref}})$ introduces a KL divergence constraint to limit the deviation of the model from the reference model.

**Difficulty-Based Curriculum Data Partitioning.** As discussed in Section 4.1.2, we partition the training data based on the difficulty of polynomial SOS decomposition in the dataset, obtaining multiple groups of training data with varying difficulty levels. Each group is then trained using a Graduated Reinforcement Policy Optimization (GRPO) approach, progressing from easier to more challenging data. Specifically, we categorize the data into three tiers based on the ascending number of polynomial variables: Polynomial-SOS pairs data with 3 to 5 variables are used in the first stage, data with 6 to 8 variables in the second stage, and the more challenging data with 9 to 10 variables in the third stage. GRPO training is conducted sequentially across these three stages.

**Reward Functions Design.** As illlstrated in Section 4.1.2, the reward function design includes three key components:

- **[accuracy reward]** encourages SOS structure conjectures with smaller errors compared to the original polynomial.

- **[format reward]** ensures that the model-generated conjectures adhere to the required SOS structure.

- **[algebraic structure penalty]** penalizes the SOS structure hypothesis based on the degree of term matching with the original polynomial.

The total reward is defined as:

$$R = w_{\text{acc}} R_{\text{acc}} + w_{\text{fmt}} R_{\text{fmt}} - P_{\text{struct}}, \tag{36}$$

where $w_{\text{acc}} + w_{\text{fmt}} = 1$, $R_{\text{acc}} \in [0, 1]$ denotes the accuracy reward based on the approximation error, $R_{\text{fmt}} \in [0, 1]$ denotes the format consistency reward, $P_{\text{struct}}$ denotes the algebraic structure penalty.

**Computation of Algebraic Structure Penalty.**  The algebraic structure penalty consists of two components: a soft penalty and a hard penalty. To accurately assess the algebraic structure, we express the original polynomial $f$ and the model-generated SOS conjecture $\hat{f}$ in terms of a common monomial basis. A coefficient threshold $\tau$ (e.g., $10^{-5}$) is set, such that a monomial is considered present in the polynomial (i.e., a nonzero term) only if the absolute value of its coefficient exceeds $\tau$.

We define Structural Deviation Rate (SDR) to quantify the discrepancy between the algebraic structure of the SOS representation and that of the original polynomial:

$$SDR = \frac{N_{\text{miss}} + N_{\text{spur}}}{N_{\text{req}}}, \tag{37}$$

where $N_{\text{req}}$ denotes the number of nonzero monomials in the original polynomial, $N_{\text{miss}}$ is the number of monomials that appear in the original polynomial but are absent in $\hat{f}$, and $N_{\text{spur}}$ is the number of **extraneous terms** present in $\hat{f}$ but not belonging to the monomial set $\mathcal{V}_f$.

The soft penalty is defined as

$$P_{\text{struct-soft}} = \lambda \cdot \min(\text{SDR}, \rho_{\max}), \tag{38}$$

where $\lambda$ controls the strength of the penalty and $\rho_{\max}$ caps the maximum penalty.

If the generated SOS structure exhibits severe algebraic inconsistencies (e.g., exceeding the degree of the original polynomial or introducing variables not present in it), a hard penalty is imposed:

$$P_{\text{struct-hard}} = \begin{cases} C_{\text{hard}}, & \text{if a structural violation occurs} \\ 0, & \text{otherwise} \end{cases} \tag{39}$$

The overall algebraic structure penalty is the sum of the soft and hard penalties:

$$P_{\text{struct}} = P_{\text{struct-hard}} + P_{\text{struct-soft}}. \tag{40}$$

## E. Further Details of Lean Verification Module

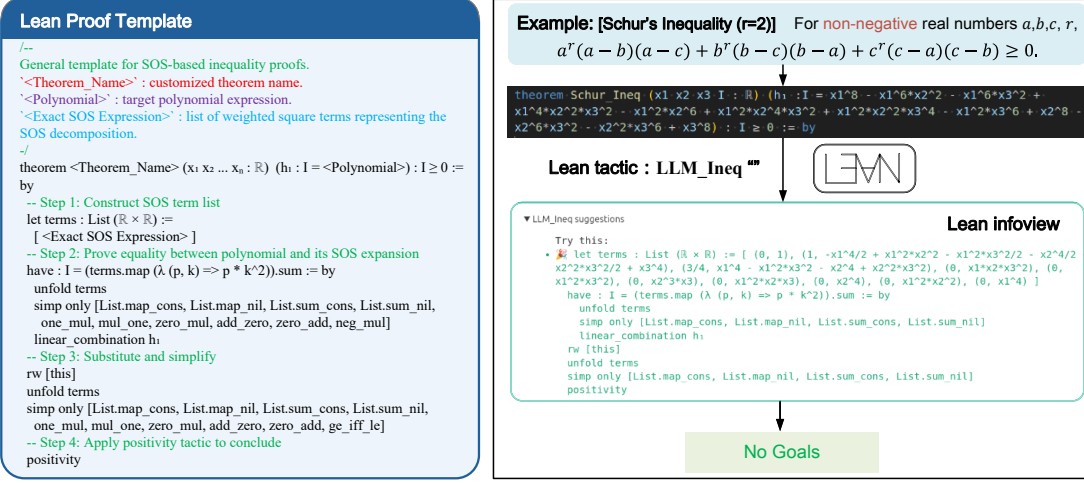

*Figure 9.* **Automated Lean Proof Generation Example. left:** The pre-defined Lean Proof Template. **right:** An Example of automated proof generation within Lean4, where the formal verification is completed using the integrated tactic *"LLM_Ineq"*.

### E.1. Lean Template of Proof

Here presents the Lean4 proof template referenced in section Section 4.3, which yields a verifiable formal proof based on the exact sum-of-squares (SOS) representation of a non-negative polynomial.

```
Lean Proof Template

/--
General template for SOS-based inequality proofs.
'<Theorem_Name>' : customized theorem name.
'<Polynomial>' : target polynomial expression.
'<Exact SOS Expression>' : list of weighted square terms representing the SOS
    decomposition.
-/
theorem <Theorem_Name> (x1 x2 ... xn : Real)
  (h1 : I = <Polynomial>) :
  I >= 0 := by
  -- Step 1: Construct SOS term list
  let terms : List (Real × Real) :=
    [ <Exact SOS Expression> ]
  -- Step 2: Prove equality between polynomial and its SOS expansion
  have : I = (terms.map (fun (p, k) => p * k^2)).sum := by
    unfold terms
    simp only [List.map_cons, List.map_nil, List.sum_cons, List.sum_nil,
      one_mul, mul_one, zero_mul, add_zero, zero_add, neg_mul]
    linear_combination h1
  -- Step 3: Substitute and simplify
  rw [this]
  unfold terms
  simp only [List.map_cons, List.map_nil, List.sum_cons, List.sum_nil,
    one_mul, mul_one, zero_mul, add_zero, zero_add, ge_iff_le]
  -- Step 4: Apply positivity tactic to conclude
  positivity
```

*Figure 10.* Lean template for inequality proof based on SOS representation.

The template uses the following placeholders:

- <Theorem_Name>: the customized theorem name;

- <Polynomial>: the target polynomial expression (often an expanded or normalized form equivalent to the original);

- <Exact SOS Expression>: a list of exact square terms constituting the SOS certificate, encoded as a Lean list of pairs (e.g., $(p, k)$) representing a squared polynomial $p$ together with its coefficient $k$, according to the chosen encoding.

The proof proceeds in four structured steps:

- **Step 1: Construct the SOS term list.** The SOS representation is introduced as terms : $\text{List}(\mathbb{R} \times \mathbb{R}) = [...]$ containing all exact squared terms required for the proof.

- **Step 2: Prove the polynomial identity.** A key lemma establishes that the target polynomial equals the sum of the expanded SOS terms. This step typically relies on simp to unfold list combinators List.map_cons, List.sum_cons, etc. and on linear_combination to derive the exact identity from the provided equality hypothesis (e.g., h1 : I = <Polynomial>) together with auxiliary algebraic equalities.

- **Step 3: Substitute and normalize the goal.** By rewriting with the established identity (rw [this]), the original goal is transformed into nonnegativity of a sum-of-squares, followed by additional simplifications to reach a form amenable to automation.

- **Step 4: Conclude via positivity automation.** Finally, the positivity tactic is invoked to prove that a sum-of-squares is nonnegative, completing the formal verification of $I(\mathbf{x}) \geq 0$.

### E.2. An Example of Automated Lean Proof Generation

Fig. 9 illustrates the structure of the proof template and an example of automated proof generation.

## F. Algorithm

Here provides the overall pseudocode for the proposed NSPI method (Algorithm 1), along with the detailed pseudocodes for the progressive two-stage training process of the neural conjecture module (Algorithm 2), the Newton iteration (Algorithm 3), and the rational recovery process (Algorithm 4) within the symbolic correction module.

---

**Algorithm 1** Overall Proof Generation Process of NSPI

---

**Input:** Target polynomial $f(x)$; SOS structure conjecturer $M$; candidate budget $K$.
**Output:** A verified Lean proof script $\mathcal{P}_{formal}$, or FAILURE.
 1: **Stage 1: Neural Conjecture**
 2: $\mathcal{S}_{approx} \leftarrow M(f(x), K)$                                          ▷ Generate $K$ SOS-structure conjectures
 3: Rank $\mathcal{S}_{approx}$ by $\theta(s) = \|\hat{f}_s(x) - f(x)\|_2$
 4: **Stage 2: Symbolic Correction**
 5: **for** each structure $s$ in $\mathcal{S}_{approx}$ **do**
 6:     $G_N \leftarrow$ NewtonIteration$(f(x), s)$            ▷ Gauss–Newton refinement to obtain $G_N$ with small backward error
 7:     $S_{rat} \leftarrow$ RationalRecovery$(G_N)$                              ▷ Interior/boundary rational recovery
 8:     **if** IsEXACTSOS$(f(x), S_{rat})$ **then**
 9:         **break**                                          ▷ An exact rational SOS certificate is found
10:     **end if**
11: **end for**
12: **Stage 3: Formal Verification**
13: **if** $S_{rat}$ is found **then**
14:     $\mathcal{P}_{formal} \leftarrow$ TEMPLATEFILL$(f(x), S_{rat})$                        ▷ Generate Lean script from templates
15:     **if** LEANCHECK$(\mathcal{P}_{formal})$ **then**
16:         **return** $\mathcal{P}_{formal}$
17:     **end if**
18: **end if**
19: **return** FAILURE.

---

**Algorithm 2** Progressive Two-Stage Training of the SOS Structure Conjecturer

---

**Input:** Base language model $M_0$; data construction methods $\mathcal{D} = \{\mathcal{D}_1, \ldots, \mathcal{D}_4\}$; curriculum schedule $\mathcal{C}$; GRPO hyperparameters.
**Output:** Trained SOS structure conjecturer $M$.
 1: **Stage 1: Data Construction and Supervised Fine-Tuning**
 2: Construct synthetic polynomial–SOS pairs $\mathcal{S} \leftarrow \bigcup_{j=1}^{4} \mathcal{D}_j$                        ▷ Four construction methods
 3: Train $M_0$ on $\mathcal{S}$ via teacher forcing to obtain $M_{\text{SFT}}$                        ▷ Supervised fine-tuning
 4: **Stage 2: Curriculum Reinforcement Learning**
 5: Evaluate $M_{\text{SFT}}$ on $\mathcal{S}$ and collect unsolved samples $\mathcal{U}$
 6: Partition $\mathcal{U}$ into curriculum buckets $\{\mathcal{U}_1, \ldots, \mathcal{U}_L\}$ by $\mathcal{C}$                        ▷ Increasing difficulty
 7: **for** $\ell = 1$ to $L$ **do**
 8:     **for** each mini-batch $\mathcal{B} \subset \mathcal{U}_\ell$ **do**
 9:         Sample SOS conjectures $\{S_i\}$ from $M$ for each $(f, S^\star) \in \mathcal{B}$                        ▷ Policy rollout
10:         Compute reward $R(S_i) \leftarrow R_{\text{acc}}(S_i) + R_{\text{fmt}}(S_i) - P_{\text{alg}}(S_i)$        ▷ Accuracy, format, and algebraic penalty
11:         Update $M$ via GRPO to maximize $\mathbb{E}[R]$                        ▷ Reinforcement learning
12:     **end for**
13: **end for**
14: **return** $M$                        ▷ Trained SOS structure conjecturer

---

---

**Algorithm 3** NEWTONITERATION: Gauss–Newton Refinement for SOS Certificates

---

**Input:** Polynomial $f(x)$; SOS structure $s$; tolerance $\tau$; max steps $T_N$.
**Output:** Refined numerical Gram matrix $G_N$ (or $\emptyset$).
 1: Build monomial basis $\mathbf{v}(x)$ implied by $s$ and initialize floating Gram matrix $G^{(0)}$.
 2: **for** $t = 0$ **to** $T_N - 1$ **do**
 3:     Compute backward error $\theta^{(t)} = \|f(x) - \mathbf{v}(x)^\top G^{(t)} \mathbf{v}(x)\|_2$.
 4:     **if** $\theta^{(t)} \leq \tau$ **then**
 5:         **return** $G_N \leftarrow G^{(t)}$
 6:     **end if**
 7:     Update $G^{(t)}$ by one Gauss–Newton step (via Cholesky parameterization).
 8: **end for**
 9: **return** $\emptyset$.

---

**Algorithm 4** RATIONALRECOVERY: Exact Rational SOS Certificate Recovery

---

**Input:** Polynomial $f(x)$; SOS structure $s$; refined Gram matrix $G_N$; rank threshold $\epsilon$; precision $\rho$.
**Output:** Exact rational SOS certificate $S_{rat}$ (or $\emptyset$).
 1: Form affine constraints from coefficient matching: $f(x) = \mathbf{v}(x)^\top G \mathbf{v}(x)$.
 2: **if** NUMRANK$(G_N, \epsilon)$ is full **then**
 3:     Project $G_N$ onto the affine constraint and rationalize entries with precision $\rho$.
 4: **else**
 5:     Perform truncated $LDL^\top$ and LLL-based rational vector recovery to preserve rank.
 6: **end if**
 7: Construct $S_{rat}$ from the recovered rational Gram matrix and **return** $S_{rat}$.

---

# G. Experimental Details

## G.1. Implementation Details

During the data construction phase of the neural conjecture module, over one million training instances were generated based on the proposed construction methods. Qwen3-8B (Team, 2025) was employed as the base model for SOS structure conjecturing. In the reinforcement learning phase, progressive multi-round training was conducted using curriculum data organized by the increasing number of variables: Stage 1 (3–5 variables), Stage 2 (6–8 variables), and Stage 3 (9–10 variables). The reward function parameters were configured with $\alpha = 0.5$, $\lambda = 0.5$, and $C_{hard} = 0.5$, with the accuracy weight $w_{acc}$ set to 0.9. Within the symbolic correction module, the tolerance threshold $\tau$ for Newton iteration was predefined as 1e-15. During the inference phase, the computational budget $k$ is set to 32, with a maximum time limit of 1 hour.

We report the computational cost of synthetic data construction, SFT, and GRPO training in Table 3. Notably, the computational cost of the training stage represents a one-time investment: once the model is trained, it can be directly deployed for inference without retraining on each new problem instance. Furthermore, additional acceleration is readily achievable, as the training process scales approximately linearly with the number of GPUs.

*Table 3.* Computational cost of each stage in NSPI.

| Stage | Hardware | Time |
|---|---|---|
| Data Generation | 2 × AMD EPYC 7H12 (200 threads) | 2 hours |
| Supervised Fine-Tuning (SFT) | 4 × NVIDIA L40S (48GB) | 7 days |
| Reinforcement Learning (GRPO) | 4 × NVIDIA L40S (48GB) | 3–4 days |

## G.2. Baseline Setups

To ensure a fair comparison, we evaluate our method against several categories of approaches, including symbol-based methods, LLM-based provers, general-purpose LLMs, and hybrid systems. Detailed configurations and specifications for each category are provided below.

- **Symbol-based Method.** We evaluate the symbolic computation tools Maple (Heck & Koepf, 1993) and Z3 (De Moura & Bjørner, 2008). For Maple, we perform symbolic decomposition and sampling of the semi-algebraic set via `SamplePoints` to check for the absence of negative values. For Z3, we employ its **SMT** solver to formulate the problem as a satisfiability task under real arithmetic.

- **LLM-based Prover.** We consider state-of-the-art (SOTA) LLM-driven Lean automated theorem provers, including DeepSeek-Prover-V2-7B (Ren et al., 2025), Goedel-Prover-V2-8B (Lin et al., 2025b), and Kimina Prover (Wang et al., 2025). Each prover is tested using its respective default prompt templates. In our main experiments, we compare their performance under a computational budget of pass@32.

- **General-purpose LLM.** This group encompasses the latest closed-source foundation models, including GPT-5.2, Gemini-3-Pro-Preview, and DeepSeek-V3.2 (DeepSeek-AI, 2025). We assess their ability to generate Lean proofs directly from Lean theorem statements.

- **Hybrid system.** We compare against the recent hybrid inequality-proving system LIPS (Li et al., 2025b), using its default experimental configuration. LIPS (Li et al., 2025b) categorizes inequality-solving techniques into two strategies, namely scaling and rewriting, handled respectively by symbolic methods and LLMs, enabling more efficient proof search. It is worth noting that a comparison with several hybrid systems (Wei et al., 2024; Li et al., 2026) was not feasible, as their source code is not publicly available for replication. Furthermore, these methods primarily focus on addressing problems with 3 or 4 variables, as evidenced by the datasets described in their respective papers.

### G.3. Details of Benchmark

Here we provide additional details on **PolyIneqBench**, the challenging inequality proving benchmark constructed in this work.

Specifically, PolyIneqBench consists of a total of **522** inequality problems, which primarily categorized into two subsets based on their origin and complexity:

- **PolyIneq-Real**: This subset contains 102 inequality problems sourced from real-world domains. It comprises two main components:

  - Classical benchmark inequalities, including the Schur inequality, Robinson polynomial, Delzell polynomial, Peyri–Parrilo polynomial, Lax polynomial, Nesbitt inequality, and Voronoi polynomial.
  - Problems derived from national and international mathematics competitions, as well as standard competition textbooks.

  Representative examples of PolyIneq-Real are presented in Table 4.

- **PolyIneq-Synth**: To extend the benchmark to higher-dimensional settings, we constructed this synthetic subset containing 420 inequalities spanning four to ten variables (with 60 problems per variable count). Each generated problem was validated by domain experts through manual quality assessment on randomly sampled instances (see Section D.2), ensuring non-triviality and challenge. Representative examples are provided in Table 5.

**Expanding to High-Dimensional Settings:** As shown in Table 4, most problems in PolyIneq-Real involve only **three** or **four** variables. PolyIneq-Synth was specifically designed to address this limitation by introducing systematically constructed problems in higher-dimensional settings.

Table 6 presents the distribution of polynomial inequalities within the PolyIneqBench, categorized by the number of variables ($n$) and their respective sample sizes.

*Table 4.* Part of PolyIneq-Real Benchmark

| Source | #Variables (n) | #Degree | #Terms | $f(x)$ |
|---|---|---|---|---|
| (Chen, 2014) | 3 | 2 | 6 | $x_1^2 - x_1 x_2 - x_1 x_3 + x_2^2 - x_2 x_3 + x_3^2$ |
| (Chen, 2014) | 3 | 6 | 6 | $x_1^6 - x_1^4 x_2^2 - x_1^2 x_3^4 + x_2^6 - x_2^4 x_3^2 + x_3^6$ |
| (Tung, 2012) | 3 | 8 | 7 | $4x_1^8 - 4x_1^2 x_2^2 - 4x_1^2 x_3^2 + 4x_2^8 - 4x_2^2 x_3^2 + 4x_3^8 + 3$ |
| (Tung, 2012) | 3 | 4 | 6 | $-2x_1^4 + 2x_1^2 x_2^2 + 2x_1^2 x_3^2 + x_2^4 - 4x_2^2 x_3^2 + x_3^4$ |
| (Riasat, 2008) | 3 | 2 | 6 | $2x_1^2 - 2x_1 x_2 - 2x_1 x_3 + 2x_2^2 - 2x_2 x_3 + 2x_3^2$ |
| IMO Short List 1998 | 3 | 8 | 7 | $4x_1^8 + 4x_1^6 + 4x_2^8 + 4x_2^6 + 4x_3^8 + 4x_3^6 - 3$ |
| (Manfrino et al., 2010) | 3 | 8 | 6 | $x_1^4 x_2^4 - x_1^4 x_2^2 x_3^2 + x_1^4 x_3^4 - x_1^2 x_2^4 x_3^2 - x_1^2 x_2^2 x_3^4 + x_2^4 x_3^4$ |
| Canada 2002 | 3 | 8 | 6 | $x_1^8 - x_1^4 x_2^2 x_3^2 - x_1^2 x_2^4 x_3^2 - x_1^2 x_2^2 x_3^4 + x_2^8 + x_3^8$ |
| Spain 1996 | 3 | 4 | 6 | $x_1^4 - 4x_1^2 x_2^2 + 2x_1^2 x_3^2 + 4x_2^4 - 4x_2^2 x_3^2 + x_3^4$ |
| Schur's Inequality ($r = 2$) | 3 | 8 | 12 | $x_1^8 - x_1^6 x_2^2 - x_1^6 x_3^2 + x_1^4 x_2^2 x_3^2 - x_1^2 x_2^6 + x_1^2 x_2^4 x_3^2 + x_1^2 x_2^2 x_3^4 - x_1^2 x_3^6 + x_2^8 - x_2^6 x_3^2 - x_2^2 x_3^6 + x_3^8$ |
| Robinson Polynomial | 3 | 6 | 10 | $x_3^6 - x_2^4 x_3^2 - x_2^2 x_3^4 + x_2^6 - x_1^2 x_3^4 + 3x_1^2 x_2^2 x_3^2 - x_1^2 x_2^4 - x_1^4 x_3^2 - x_1^4 x_2^2 + x_1^6$ |
| (Mildorf, 2005) | 3 | 4 | 9 | $x_1^4 - 3x_1^3 x_2 + 2x_1^2 x_2^2 + 2x_1^2 x_3^2 - 3x_1 x_3^3 + x_2^4 - 3x_2^3 x_3 + 2x_2^2 x_3^2 + x_3^4$ |
| (Parvardi, 2011) | 3 | 8 | 12 | $x_1^8 + 4x_1^6 x_2^2 - 4x_1^6 x_3^2 + 2x_1^4 x_2^4 + 2x_1^4 x_3^4 - 4x_1^2 x_2^6 + 4x_1^2 x_3^6 + x_2^8 + 4x_2^6 x_3^2 + 2x_2^4 x_3^4 - 4x_2^2 x_3^6 + x_3^8$ |
| (Tung & Zhe, 2014) | 3 | 6 | 10 | $4x_1^6 + 12x_1^4 x_2^2 - 15x_1^4 x_3^2 - 15x_1^2 x_2^4 - 3x_1^2 x_2^2 x_3^2 + 12x_1^2 x_3^4 + 4x_2^6 + 12x_2^4 x_3^2 - 15x_2^2 x_3^4 + 4x_3^6$ |
| (Lee, 2005) | 3 | 6 | 10 | $x_1^2 x_2^2 x_3^2 - 2x_1^2 x_2 x_3 + x_1^2 - 2x_1 x_2^2 x_3 - 2x_1 x_2 x_3^2 + 2x_1 x_2 + 2x_1 x_3 + x_2^2 + 2x_2 x_3 + x_3^2$ |
| (Andreescu, 2019) | 4 | 6 | 15 | $x_1^6 + 3x_1^4 x_2^2 + 3x_1^2 x_2^4 - 4x_1^2 x_2^2 - 4x_1^2 x_3^2 - 4x_1^2 x_4^2 + x_2^6 - 4x_2^2 x_3^2 - 4x_2^2 x_4^2 + x_3^6 + 3x_3^4 x_4^2 + 3x_3^2 x_4^4 - 4x_3^2 x_4^2 + x_4^6 + 8$ |
| (El Din, 2008) | 4 | 8 | 20 | $1 - 8x_3^2 x_4^2 - 196608 x_5^3 x_1^2 x_4^2 x_3 + 1536 x_5 x_1 x_4^4 x_3^2 + 21504 x_5^2 x_1 x_4^2 x_3 - 4096 x_5^2 x_1 x_3^3 x_4^2 - 384 x_5 x_1 x_4^2 + 1024 x_5^2 x_1 x_3 + 16x_3^4 x_4^4 - 72x_3^2 x_4^4 + 1024 x_3^2 x_5^2 + 36864 x_5^2 x_1^2 x_4^4 - 3456 x_5 x_1 x_4^4 + 262144 x_5^4 x_1^2 x_3^2 - 32768 x_5^3 x_1 x_3^2 + 256 x_3^3 x_4^2 x_5 - 576 x_3 x_5 x_4^2 + 81x_4^4 + 64x_3 x_5 - 18x_4^2$ |
| (Magron et al., 2023) | 3 | 4 | 9 | $x_1^4 + x_1 x_2^3 + x_2^4 - 3x_1^2 x_2 x_3 - 4x_1 x_2^2 x_3 + 2x_1^2 x_3^2 + x_1 x_3^3 + x_2 x_3^3 + x_3^4$ |

*Table 5.* Part of PolyIneq-Synth Benchmark

| #Variables (n) | #Degree | #Terms | $f(x)$ |
|---|---|---|---|
| 4 | 12 | 7 | $6x_1^{12} - 2x_1^6 x_2 x_3^3 - 4x_1^6 x_4^2 x_3^4 + 5x_2^2 x_4^6 + 4x_2^2 x_3^6 - 2x_2 x_4^5 x_3^4 + 3x_4^4 x_3^8$ |
| 5 | 12 | 10 | $8x_1^{12} + 2x_1^6 x_2 x_3 x_4 + 2x_1^6 x_5^2 + 8x_2^2 x_3^{10} + 2x_2^2 x_3^6 x_4 + 6x_2^2 x_3^2 x_4^2 - 2x_2 x_3 x_4 x_5^2 + 7x_4^2 x_5^6 - 2x_4 x_5^5 + 8x_5^4$ |
| 6 | 12 | 13 | $5x_1^{12} + 2x_1^6 x_2^6 + 2x_1^6 x_3^3 x_4^2 + 8x_2^{12} - 2x_2^6 x_3^3 x_4^2 + 2x_2^6 x_3^2 x_4^2 x_5^2 + 2x_2^6 x_4 x_6^5 + 5x_3^6 x_4^4 + 2x_3^5 x_4^4 x_5^2 + 5x_3^4 x_4^4 x_5^4 - 2x_3^3 x_4^3 x_5^5 - 2x_3^2 x_4^3 x_5^5 x_6^2 + 7x_4^2 x_6^{10}$ |
| 7 | 12 | 14 | $4x_1^{10} x_2^2 + 4x_1^5 x_3^3 x_2^2 x_4^2 - 2x_1^5 x_2 x_3^3 x_5 - 6x_1^5 x_2 x_6^2 x_7^3 + 2x_1^5 x_2 x_6 x_4^2 x_7^3 + 5x_2^4 x_3^4 x_4^4 - 10x_2^2 x_3^5 x_5 x_4^2 - 2x_2^2 x_3^3 x_6^3 x_4^2 x_7^3 + 4x_2^2 x_3^2 x_6 x_4^4 x_7^3 + 9x_3^6 x_5^2 - 8x_3^3 x_5 x_6^3 x_7^3 - 4x_3^3 x_5 x_6 x_4^2 x_7^3 + 8x_6^6 x_7^6 + 2x_2^2 x_4^4 x_7^6$ |
| 8 | 16 | 23 | $9x_1^{12} x_7^2 + 22x_1^{10} x_3^2 x_5^2 x_8^2 + 16x_1^6 x_3^2 x_5^2 x_6 x_7 x_8^2 - 44x_1^6 x_3^2 x_5 x_8 + 4x_1^6 x_4^2 x_5^4 + 6x_1^5 x_3^2 x_5^2 x_6 x_8 + 8x_1^5 x_3^2 x_5 x_7 x_8 - 8x_1^4 x_3 x_4 x_5^4 x_6 x_7 x_8 + 11x_1^4 x_4^2 x_5^4 x_8^2 - 16x_1^3 x_3 x_4 x_5^3 x_6 + 8x_1^3 x_3 x_4 x_5^4 x_7 + 23x_1^2 x_3^2 x_5^2 x_6^2 x_7^2 x_8^2 - 32x_1^2 x_3^2 x_5 x_6 x_7 x_8 + 28x_1^2 x_3^2 + 32x_1 x_3^2 x_5^2 x_6^2 x_7 x_8 - 8x_1 x_3^2 x_5 x_6 x_7^2 x_8 - 16x_1 x_3^2 x_5 x_6 - 8x_1 x_3^2 x_7 + 2x_4^2 x_4^2 x_5^2 x_6^2 x_7^2 x_8^2 + 4x_2^2 x_3 x_4 x_5 x_6 x_7^2 x_8 + 21x_3^2 x_5^2 x_6^2 - 16x_3^2 x_5 x_6 x_7 + 8x_3^2 x_7^2$ |
| 9 | 18 | 34 | $14x_1^{12} x_3^2 x_9^2 - 2x_1^{12} x_3 x_9 + 13x_1^{12} + 2x_1^8 x_2 x_3 x_5 x_9^2 - 2x_1^8 x_2 x_5 x_9 - 2x_1^7 x_2^2 x_3^2 x_4 x_5 x_6^2 - 4x_1^7 x_2 x_3^2 x_5^2 x_6 x_7 x_8 x_9 + 2x_1^7 x_2^2 x_9 + 2x_1^6 x_3^3 x_6^2 x_9 + 2x_1^6 x_3^2 x_5 x_8 x_9 + 2x_1^6 x_3^2 x_6^2 + 4x_1^6 x_3 x_4 x_5 x_8 x_9^2 + 2x_1^6 x_3 x_5 x_8 - 2x_1^6 x_7 x_8 x_9^4 + 15x_1^4 x_2^2 x_5^2 x_9^2 + 2x_1^3 x_2^2 x_3 x_5^3 x_6 x_7 x_8 x_9 - 2x_1^3 x_2 x_5^3 x_9^2 + 11x_1^2 x_2^4 x_3^4 x_4^2 x_5^2 x_6^4 + 12x_1^2 x_2^2 x_3^2 x_5^4 x_6^2 x_7^2 x_8^2 - 4x_1^2 x_2 x_3 x_5^2 x_8 x_9 - 2x_1^2 x_2 x_4 x_5^2 x_8 x_9^2 + 2x_1^2 x_2 x_5 x_7 x_8 x_9^5 + 13x_1^2 x_5^4 x_9^2 - 2x_1 x_2^2 x_3^3 x_4 x_5^2 x_6^2 x_8 - 2x_1 x_2^2 x_3^2 x_4 x_5 x_6^2 x_7 x_8 x_9^4 - 2x_1 x_2 x_3^2 x_5^3 x_6 x_7 x_8^2 + 2x_1 x_3^2 x_5^2 x_6^2 x_9 + 2x_1 x_4 x_5^3 x_8 x_9 + 16x_3^4 x_6^4 - 2x_3^3 x_5 x_6^2 x_8 + 13x_3^2 x_5^2 x_8^2 - 2x_3 x_4 x_5^2 x_8^2 x_9 + 12x_4^2 x_5^2 x_8^2 x_9 + 14x_7^2 x_8^2 x_9^8$ |
| 10 | 18 | 42 | $10x_1^{10} x_4^6 + 2x_1^7 x_4^3 x_7^3 x_8 - 2x_1^6 x_3 x_4^4 x_5 x_6 x_7 - 4x_1^6 x_3 x_4^3 x_5 x_9 - 2x_1^5 x_2 x_4^3 x_6^4 x_7^2 - 2x_1^5 x_4^3 x_5^2 x_8^3 x_9 + 10x_1^4 x_7^6 x_8^2 + 2x_1^3 x_2 x_3 x_6 x_7^5 x_8^2 x_9 + 4x_1^3 x_3 x_4 x_5 x_6 x_7^4 x_8 + 14x_1^2 x_2^2 x_3^2 x_6^2 x_7^2 x_8^2 x_9^2 - 4x_1^2 x_2 x_3^2 x_5 x_6 x_7^2 x_8 x_9 - 2x_1^2 x_2 x_3 x_7^3 x_8 - 2x_1^2 x_2 x_6^2 x_7^5 x_8 + 2x_1^2 x_2 x_7^3 x_8 + 15x_1^2 x_3^2 x_4^2 x_5^2 x_6^2 x_7^2 - 4x_1^2 x_3^2 x_4 x_5^2 x_6 x_7 x_9 + 16x_1^2 x_3^2 x_5^2 x_9^2 + 2x_1 x_{10} x_2^3 x_3^2 x_4 x_6 x_7^3 x_8^2 x_9 + 2x_1 x_{10} x_2^2 x_3^2 x_4 x_5 x_7 x_8 x_9^2 - 2x_1 x_2^2 x_3^2 x_6 x_7^2 x_8 x_9 - 2x_1 x_2^2 x_3 x_6^7 x_7^4 x_8 x_9 - 2x_1 x_2^2 x_3 x_6 x_7^2 x_8 x_9 - 2x_1 x_2 x_3^2 x_5 x_9 - 4x_1 x_2 x_3 x_4 x_5 x_6^7 x_7^3 + 2x_1 x_2 x_3 x_4 x_5 x_6 x_7 - 4x_1 x_2 x_3 x_5^2 x_6 x_7^2 x_8^4 x_9 - 2x_1 x_2 x_3 x_5 x_6^6 x_7^2 x_9 - 2x_1 x_2 x_3 x_5 x_9 + 2x_1 x_3 x_5^3 x_8^3 x_9 + 13x_{10}^2 x_2^4 x_3^2 x_4^2 x_7^2 x_8^2 x_9^2 + 2x_{10} x_2^3 x_3^2 x_4 x_7 x_8 x_9 - 2x_{10} x_2^3 x_3 x_4 x_6^6 x_7^3 x_8 x_9 + 2x_{10} x_2^3 x_3 x_4 x_7 x_8 x_9 - 2x_{10} x_2^2 x_3 x_4 x_5^2 x_7 x_8^4 x_9 + 10x_2^2 x_3^2 - 2x_2^2 x_3 + 14x_2^2 x_6^{12} x_7^4 + 2x_2^2 x_6^6 x_7^2 + 15x_2^2 + 2x_2 x_3 x_5^2 x_8^3 x_9 - 4x_2 x_5^2 x_8^3 x_9 + 12x_5^4 x_8^6 x_9^2$ |

# H. Case Study

Here we provide detailed case studies illustrating the complete proof process of NSPI on two representative examples drawn from PolyIneqBench.

Case 1 is a classical inequality from the PolyIneq-Real subset, whereas Case 2 is a challenging competition-level synthetic inequality from the PolyIneq-Synth subset, corresponding to a higher-dimensional and more difficult setting. Notably, for Case 2, NSPI is the **only method** among all baselines that successfully completes the proof.

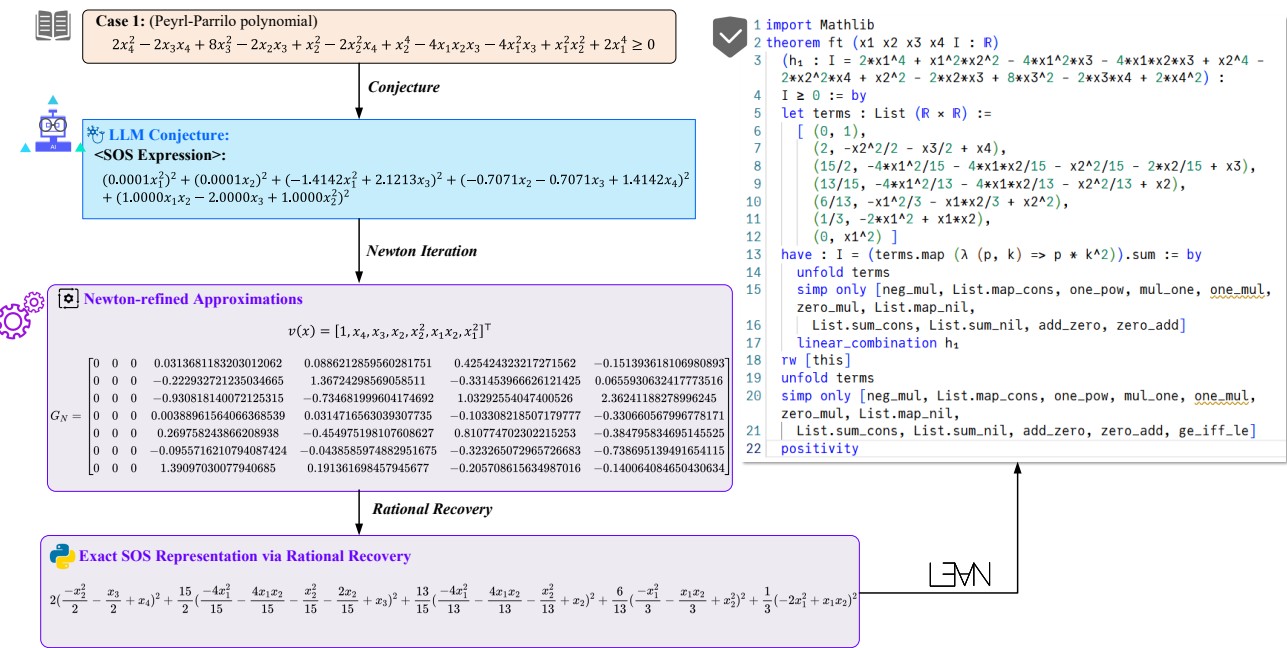

*Figure 11.* **Detailed workflow of the NSPI method applied to the Peyrl-Parrilo polynomial from PolyIneq-Real (Case 1).** This example illustrates the complete pipeline from initial conjecture to formal certification.

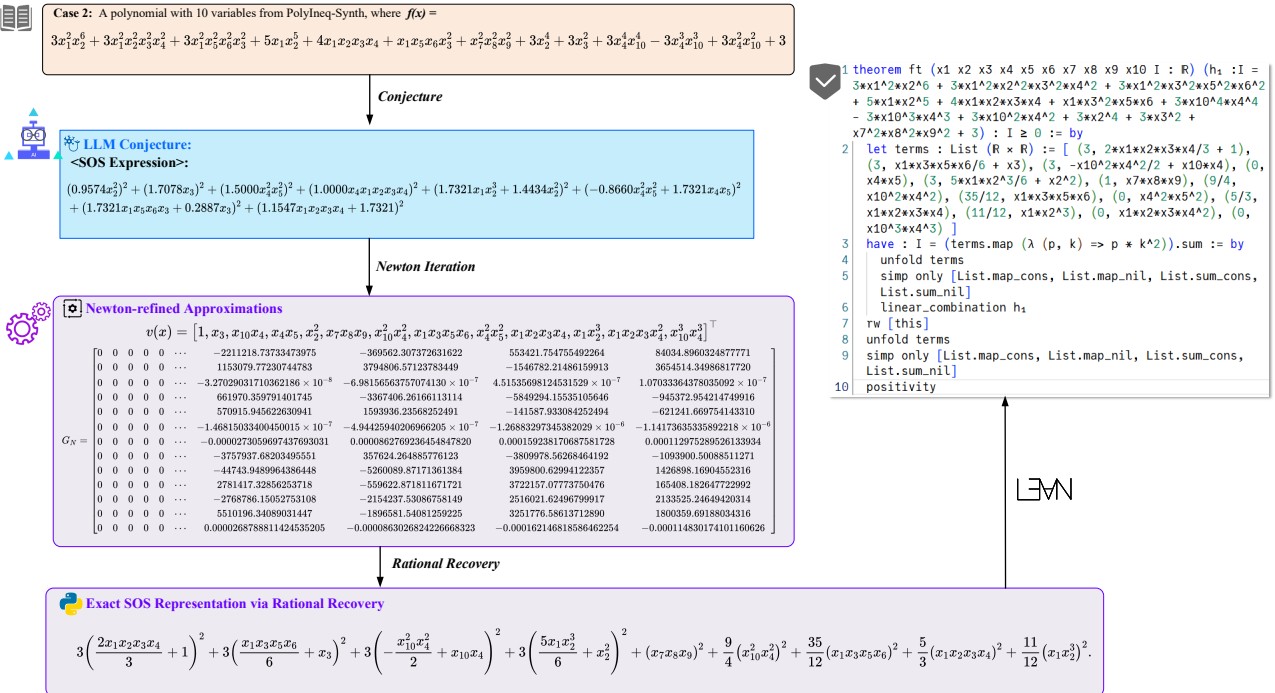

*Figure 12.* **Case study of a challenging synthetic polynomial from PolyIneq-Synth.** This 10-variable instance represents a significant challenge in high-dimensional formal reasoning; notably, **NSPI is the sole method among all evaluated baselines (Case 2)** capable of successfully generating a verified formal proof.

*Table 6.* Distribution of Polynomial Inequalities by Variable Count in PolyIneqBench.

| Variable Count (n) | Sample Size |
|:---:|:---:|
| 3 | 84 |
| 4 | 77 |
| 5 | 60 |
| 6 | 61 |
| 7 | 60 |
| 8 | 60 |
| 9 | 60 |
| 10 | 60 |

### H.1. Case 1: Peyrl-Parrilo polynomial.

Case 1 features a classic non-negative polynomial drawn from (Peyrl & Parrilo, 2008b; Le & Van Barel, 2014). Fig. 11 presents an example of generating a complete proof using the NSPI method.

### H.2. Case 2: A challenging synthetic polynomial.

We present a challenging 10-variable synthetic polynomial instance from the PolyIneq-Synth dataset. Notably, for this specific case, NSPI is the only method among all considered baselines that successfully generates a complete formal proof.

## I. More Experimental Results

### I.1. Results under Different Computational Budgets

Here, we report the performance of various LLM-based provers and our NSPI method under different computational budgets. The results on PolyIneqBench are summarized in Table 7.

*Table 7.* Comparative results of multiple methods on PolyIneqBench under different computational budgets

| Method Name | Computational Budget | PolyIneq-Real | PolyIneq-Synth |
|:---|:---:|:---:|:---:|
| DeepSeek-Prover-V2 | pass@8
pass@16
pass@32 | 28.43%
32.35%
37.25% | 0.00%
0.00%
0.00% |
| Goedel-Prover-V2 | pass@8
pass@16
pass@32 | 15.69%
16.67%
18.63% | 0.48%
0.48%
0.48% |
| Kimina-Prover | pass@8
pass@16
pass@32 | 27.45%
30.39%
33.33% | 0.24%
0.24%
0.71% |
| **NSPI (ours)** | pass@8
pass@16
pass@32 | 36.27%
40.20%
43.14% | 23.57%
25.48%
25.95% |

## I.2. Detailed Comparative Study on Benchmark Difficulty Metrics

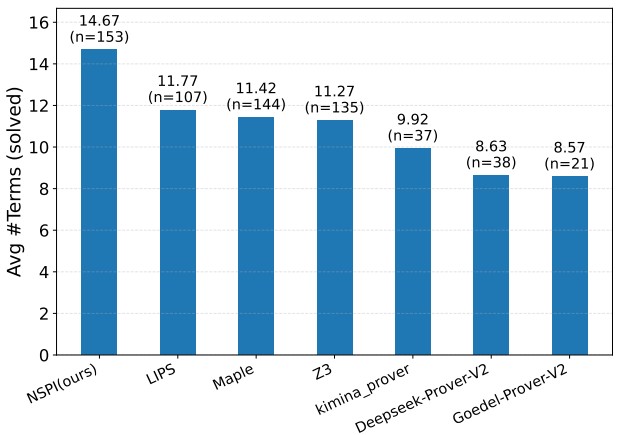

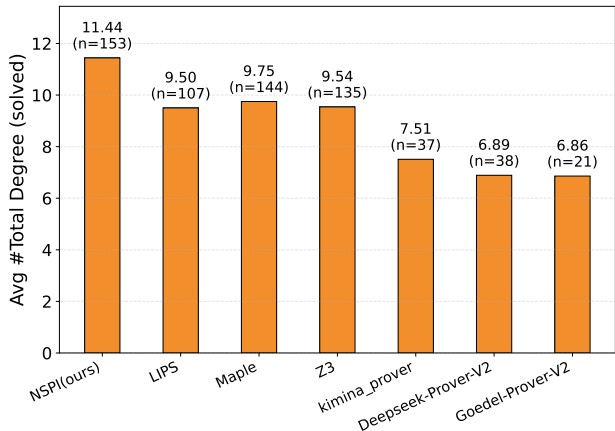

*(a)* Average Number of Terms in Polynomials Solved by Different Methods.

*(b)* Average Total Degree of Terms in Polynomials Solved by Different Methods.

*Figure 13.* Comparative analysis of different polynomial solving methods: (a) average number of terms; (b) average total degree.

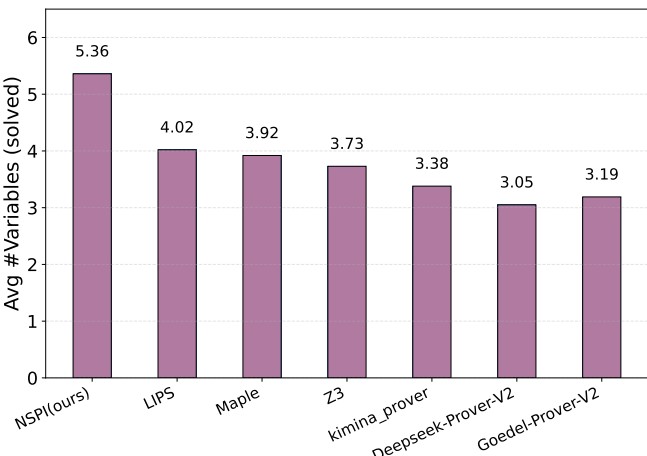

*Figure 14.* **Average number of variables in polynomial inequalities successfully verified by each method.** This metric illustrates the scalability of various provers, where NSPI (ours) demonstrates a superior capability in handling higher-dimensional problems compared to both symbolic baselines and LLM-based provers.

Fig. 13 and Fig. 14 present a comparative analysis of different polynomial inequality solving methods in terms of average number of terms (Fig. 13 (a)), average total degree (Fig. 13 (b)), and average number of variables (Fig. 14) in successfully solved instances.

Fig. 13 shows that our NSPI outperforms other methods in terms of the average number of terms and total degree in the polynomials it solves, suggesting its ability to handle more complex polynomials.

Fig. 14 highlights that NSPI also excels in handling higher-dimensional problems, as it successfully solves polynomials with the largest average number of variables, outperforming symbolic and LLM-based provers. These metrics together demonstrate NSPI's superior scalability in solving complex and high-dimensional polynomial inequalities compared to both symbolic and LLM-based approaches.

## J. Failure Mode Analysis

To better understand the performance boundaries of NSPI, we conduct a systematic analysis of failed instances on PolyIneqBench and identify two primary failure modes.

**Failure Mode 1: Absence of SOS Decomposition.** One source of failure arises from the absence of an SOS decomposition for the target polynomial itself. For such instances, natural proof strategies typically rely on non-SOS techniques, including AM-GM transformations, variable substitutions, and classical inequality templates. These failures therefore reflect the applicability boundary of SOS-based methods rather than deficiencies in the reliability of the proving pipeline itself. An example is given below: $x_1^2 x_2^2 x_3^2 + x_1^2 - 2x_1 x_2 - 2x_1 x_3 + x_2^2 - 2x_2 x_3 + x_3^2 + 2$. The polynomial was verified by YALMIP to admit no SOS representation. We verified all instances using the SDP solver and found that approximately 27 instances in PolyIneq-Real admit no feasible SOS certificates.

**Failure Mode 2: Insufficient LLM Conjecture Quality.** For the remaining failed instances, the approximate SOS conjectures generated by the LLM lacked sufficient accuracy, preventing the symbolic correction module from recovering exact certificates. Typical failure patterns include missing monomials, redundant terms, or overly large coefficient deviations in the proposed SOS structure, causing the Gauss–Newton refinement procedure to fail to converge to a valid rational solution. A representative example is presented below, including the input polynomial, the LLM-generated conjecture, its expanded form, and the residual monomials that cannot be matched. Approximately 31 instances in PolyIneq-Real fall into this category.

---

**Failure Mode 2: Insufficient LLM Conjecture Quality**

**Input polynomial:**

$$x_1^8 x_3^4 - 2x_1^6 x_2^4 x_3^2 + x_1^6 x_2^2 x_3^4 + x_1^4 x_2^8 + x_1^4 x_2^6 x_3^2 - 2x_1^4 x_2^2 x_3^6 - 2x_1^2 x_2^6 x_3^4 + x_1^2 x_2^4 x_3^6 + x_2^4 x_3^8$$

**LLM-generated conjecture:**

$$\left(x_1^4 x_3^2 - x_1^2 x_2^4 - 0.5 x_2^2 x_3^4\right)^2 + \left(x_1^3 x_2 x_3^2 - x_1 x_2^2 x_3^3\right)^2 + \left(0.8660 x_1^2 x_2^3 x_3 - 0.8660 x_2^2 x_3^4\right)^2$$

**Expanded form:**

$$x_1^8 x_3^4 - 2x_1^6 x_2^4 x_3^2 + x_1^6 x_2^2 x_3^4 + x_1^4 x_2^8 + 0.749956 x_1^4 x_2^6 x_3^2 - 2x_1^4 x_2^3 x_3^5$$
$$- x_1^4 x_2^2 x_3^6 + x_1^2 x_2^6 x_3^4 - 1.499912 x_1^2 x_2^5 x_3^5 + x_1^2 x_2^4 x_3^6 + 0.999956 x_2^4 x_3^8$$

Two **extra** monomials are introduced:

$$x_1^4 x_2^3 x_3^5, \quad x_1^2 x_2^5 x_3^5$$

---

Together, these two failure modes account for the vast majority of unresolved instances in PolyIneqBench, highlighting two complementary directions for future improvement. The first direction involves expanding the framework beyond polynomial SOS representations, such as by incorporating Positivstellensatz or rational function SOS decompositions. The second direction focuses on enhancing the quality of LLM-generated conjectures through enriched training datasets or stronger reasoning scaffolding.

# K. Extended Discussion

Neuro-symbolic reasoning systems represent a highly valuable and promising research direction. It is worth noting that the neuro-symbolic approach to inequality proving proposed in this work differs fundamentally from existing neuro-symbolic practices. In particular, we innovatively explore the feasibility of employing large language models (LLMs) as *core symbolic conjecture engines* that directly provide end-to-end symbolic priors, thereby opening a promising new direction in which LLMs function as the central component for symbolic conjecturing. Moreover, we extend formalized theorem proving for polynomial inequalities to challenging high-dimensional cases with up to 10 variables, thereby *extending the boundary of automated proof generation for polynomial inequalities*. To further contextualize the contributions of this work, we provide a comparative discussion with existing neuro-symbolic reasoning approaches.

### K.1. A New Paradigm: LLMs as Core Symbolic Conjecture Engines

In recent years, several neuro-symbolic hybrid reasoning approaches have demonstrated encouraging progress in automated theorem proving. However, in most existing systems, neural modules primarily serve *auxiliary* roles that guide or accelerate symbolic derivation processes by reducing the search space.

For example, AlphaGeometry (Trinh et al., 2024) uses an LLM to predict auxiliary constructions, while the proof is completed by a symbolic solver that enumerates derivation rules. AIPS (Wei et al., 2024) employs a learned value network to evaluate intermediate subgoals during symbolic proof search. LIPS (Li et al., 2025b) leverages an LLM to generate candidate goal-rewriting steps and selects promising subgoals to prune the strategy space. Similarly, IneqSearch (Li et al., 2026) leverages LLMs to perform inequality transformations or to select appropriate transformation rules, integrating an iterative learning mechanism. In these systems, neural components assist decision-making within a symbolic engine, whereas the ability of neural models to directly provide symbolic reasoning content remains relatively underexplored.

In contrast, our work treats the LLM as a *core symbolic conjecturing engine*. The LLM directly produces the symbolic priors required to complete a proof, in the form of SOS structure conjectures. These conjectures are subsequently refined through Newton-style numerical refinement and rational recovery (Section 4.2), and finally transformed into complete formal Lean proofs (Section 4.3). This end-to-end pipeline demonstrates the feasibility of a new neuro-symbolic paradigm in which LLMs provide high-level symbolic structure, while symbolic computation and formal verification ensure correctness.

### K.2. Extending the Frontier of Formal Polynomial Inequality Proving

Automated formal proving of polynomial inequalities remains a challenging problem. Although automated theorem-proving techniques have advanced rapidly, automated proving of polynomial inequalities continue to pose significant difficulties, especially in settings involving many variables or high-dimensional structures.

Learning-based approaches to formal inequality proving often face a bottleneck in the availability of high-quality formal data. Widely used theorem-proving benchmarks, such as miniF2F (Zheng et al., 2021), ProofNet (Azerbayev et al., 2023) and Putnum (Tsoukalas et al., 2024) contain very few polynomial inequality problems; this scarcity is particularly pronounced for high-dimensional multivariate polynomials. Existing neuro-symbolic inequality proving methods also tend to focus on low-dimensional cases with a small number of variables, and struggle to scale to more complex settings.

The neuro-symbolic approach proposed in this work extends Lean-based formal polynomial inequality proving to settings involving up to ten variables, and further introduces a challenging benchmark covering inequalities with three to ten variables. As demonstrated by the results in Section 5, our method consistently outperforms all LLM-based and symbolic baselines on this challenging benchmark, with particularly strong performance in cases involving a larger number of variables. These results indicate that the proposed approach significantly expands the frontier of formal polynomial inequality theorem proving.

## L. More Examples

Here we provide additional examples that were successfully solved exclusively by our proposed NSPI method among all baseline approaches.

---

**NSPI Achieves Solution While Baseline Methods Fail(1)**

**Polynomial:**

$8x_1^6x_2^2x_3^2x_4^2 + 6x_1^4x_5^2x_6x_2x_3x_4 + 8x_1^3x_7^6x_2x_3x_4 + 8x_1^3x_6x_2^3x_3^2x_4 + 6x_1^3x_2x_3x_4^3x_8^2x_9^2 + 9x_1^2x_5^4x_6^2 - 6x_1x_7^6x_5^2x_6 - 4x_1x_5^2x_6^2x_2^2x_3 + 10x_1x_5^2x_6x_4^2x_8^2x_9^2 + 9x_7^{12} + 2x_7^6x_6x_2^2x_3 + 4x_7^6x_4^2x_8^2x_9^2 + 9x_6^2x_2^2x_3^2 - 8x_6x_2^2x_3x_4^2x_8^2x_9^2 + 6x_4^4x_8^4x_9^4$

**Lean Proof:**

```
import Mathlib
theorem f1 (x1 x2 x3 x4 x5 x6 x7 x8 x9 I : Real) (h1 :I = 8*x1
    ^6*x2^2*x3^2*x4^2 + 6*x1^4*x2*x3*x4*x5^2*x6 + 8*x1^3*x2^3*x3^2*x4*x6 + 6*x1
    ^3*x2*x3*x4^3*x8^2*x9^2 + 8*x1^3*x2*x3*x4*x7^6 + 9*x1^2*x5^4*x6^2 - 4*x1*x2^2*x3*x5^2*x6^2 + 10*x1*x4^2*x5^2*x6*
    x8^2*x9^2 - 6*x1*x5^2*x6*x7^6 + 9*x2^4*x3^2*x6^2 - 8*x2^2*x3*x4^2*x6*x8^2*x9^2 + 2*
    x2^2*x3*x6*x7^6 + 6*x4^4*x8^4*x9^4 + 4*x4^2*x7^6*x8^2*x9^2 + 9*x7^12) : I >= 0 :=
  by
  let terms : List (Real × Real) := [ (0, 1), (0, x2^2*x3), (9, x1^3*x2*x3*x4/3 + x1*
      x5^2*x6 - 2*x2^2*x3*x6/9 + 5*x4^2*x8^2*x9^2/9 - x7^6/3), (77/9, 6*x1^3*x2*x3*x4
      /11 + x2^2*x3*x6 - 26*x4^2*x8^2*x9^2/77 + 3*x7^6/77), (173/77, 224*x1^3*x2*x3*x4
      /173 + x4^2*x8^2*x9^2 + 291*x7^6/173), (282/173, -13*x1^3*x2*x3*x4/282 + x7^6),
      (0, x3^2*x4^4), (193/282, x1^3*x2*x3*x4), (0, x4^2*x6*x8^2*x9^2), (0, x6*x7^6) ]
  have : I = (terms.map (fun (p, k) => p * k^2)).sum := by
    unfold terms
    simp only [List.map_cons, List.map_nil, List.sum_cons, List.sum_nil]
    linear_combination h1
  rw [this]
  unfold terms
  simp only [List.map_cons, List.map_nil, List.sum_cons, List.sum_nil]
  positivity
```

---

**NSPI Achieves Solution While Baseline Methods Fail(2)**

**Polynomial:**

$7x_1^2x_2^2x_3^2x_4^2 - 2x_1x_5^2x_2x_3x_4x_6x_7x_8 - 4x_1x_5x_2x_3x_4x_6x_9^4 - 6x_1x_2x_3x_4x_6^5x_8 - 10x_1x_2x_3x_4x_6x_7^3x_8^2 + 8x_5^4x_6^2x_7^2x_8^2 - 4x_5^2x_6^2x_7^4x_8^3 + 8x_5^2x_6^2x_9^8 - 4x_5x_6^6x_8x_9^4 - 2x_5x_6^2x_7^3x_8^2x_9^4 + 5x_6^{10}x_8^2 + 6x_6^6x_7^3x_8^3 + 6x_6^2x_7^6x_8^4$

**Lean Proof:**

```
import Mathlib
theorem f1 (x1 x2 x3 x4 x5 x6 x7 x8 x9 I : Real) (h1 :I = 7*x1^2*x2^2*x3^2*x4^2 - 2*x1
    *x2*x3*x4*x5^2*x6*x7*x8 - 4*x1*x2*x3*x4*x5*x6*x9^4 - 6*x1*x2*x3*x4*x6^5*x8 - 10*x1*
    x2*x3*x4*x6*x7^3*x8^2 + 8*x5^4*x6^2*x7^2*x8^2 - 4*x5^2*x6^2*x7^4*x8^3 + 8*x5^2*x6
    ^2*x9^8 - 4*x5*x6^6*x8*x9^4 - 2*x5*x6^2*x7^3*x8^2*x9^4 + 5*x6^10*x8^2 + 6*x6^6*x7
    ^3*x8^3 + 6*x6^2*x7^6*x8^4) : I >= 0 := by
  let terms : List (Real × Real) := [ (0, 1), (0, x3*x4^2), (0, x4^3*x6), (7, x1*x2*x3
      *x4 - x5^2*x6*x7*x8/7 - 2*x5*x6*x9^4/7 - 3*x6^5*x8/7 - 5*x6*x7^3*x8^2/7), (55/7,
      x5^2*x6*x7*x8 - 2*x5*x6*x9^4/55 - 3*x6^5*x8/55 - 19*x6*x7^3*x8^2/55), (0, x3^2*x4
      *x6*x7), (408/55, x5*x6*x9^4 - 79*x6^5*x8/204 - 139*x6*x7^3*x8^2/408), (257/408,
      -110*x6^5*x8/257 + x6*x7^3*x8^2), (633/257, x6^5*x8), (0, x4*x6^5), (0, x3^4*x5*
      x6), (0, x5^2*x6^2*x7^4*x8^3), (0, x6^6*x7^3*x8^3) ]
  have : I = (terms.map (fun (p, k) => p * k^2)).sum := by
    unfold terms
    simp only [List.map_cons, List.map_nil, List.sum_cons, List.sum_nil]
    linear_combination h1
  rw [this]
  unfold terms
  simp only [List.map_cons, List.map_nil, List.sum_cons, List.sum_nil]
  positivity
```

**NSPI Achieves Solution While Baseline Methods Fail(3)**

**Polynomial:**

$11x_1^4x_2^4x_3^2 - 4x_1^3x_2^3x_3 - 12x_1^3x_2^2x_3^3x_5x_6 - 6x_1^3x_2^2x_3^2x_5^5x_7 + 2x_1^2x_2^3x_3^3x_5 - 2x_1^2x_2^3x_3x_4 - 8x_1^2x_2^3x_3x_5x_7 - 2x_1^2x_2^2x_3x_7^4 + 10x_1^2x_2^2 + 4x_1^2x_2x_3^2x_5x_6 + 4x_1^2x_2x_3x_5^5x_7 + 12x_1^2x_3^4x_5^2x_6^2 + 2x_1^2x_3^3x_5^6x_6x_7 + 12x_1^2x_3^2x_5^{10}x_7^2 - 12x_1x_2^2x_3^2x_5 - 4x_1x_2^2x_4 - 2x_1x_2^2x_5x_7 - 6x_1x_2x_3^4x_5^2x_6 + 2x_1x_2x_3^3x_5^6x_7 + 8x_1x_2x_3^2x_4x_5x_6 + 2x_1x_2x_3^2x_5^2x_6x_7 + 4x_1x_2x_3x_4x_5^5x_7 - 6x_1x_2x_3x_5^6x_7^2 - 14x_1x_2x_7^4 + 10x_1x_3^2x_5x_6x_7^4 + 2x_1x_3x_5^5x_7^5 + 13x_2^2x_3^4x_5^2 + 4x_2^2x_3^2x_4x_5 - 12x_2^2x_3^2x_5^2x_7 + 10x_2^2x_4^2 + 14x_2^2x_5^2x_7^2 + 8x_2x_3^2x_5x_7^4 + 2x_2x_4x_7^4 + 4x_2x_5x_7^5 + 12x_7^8$

**Lean Proof:**

```
import Mathlib
theorem f1 (x1 x2 x3 x4 x5 x6 x7 I : Real) (h1 :I = 11*x1^4*x2^4*x3^2 - 4*x1^3*x2^3*x3
    - 12*x1^3*x2^2*x3^3*x5*x6 - 6*x1^3*x2^2*x3^2*x5^5*x7 + 2*x1^2*x2^3*x3^3*x5 - 2*x1
    ^2*x2^3*x3*x4 - 8*x1^2*x2^3*x3*x5*x7 - 2*x1^2*x2^2*x3*x7^4 + 10*x1^2*x2^2 + 4*x1^2*
    x2*x3^2*x5*x6 + 4*x1^2*x2*x3*x5^5*x7 + 12*x1^2*x3^4*x5^2*x6^2 + 2*x1^2*x3^3*x5^6*x6
    *x7 + 12*x1^2*x3^2*x5^10*x7^2 - 12*x1*x2^2*x3^2*x5 - 4*x1*x2^2*x4 - 2*x1*x2^2*x5*x7
    - 6*x1*x2*x3^4*x5^2*x6 + 2*x1*x2*x3^3*x5^6*x7 + 8*x1*x2*x3^2*x4*x5*x6 + 2*x1*x2*x3
    ^2*x5^2*x6*x7 + 4*x1*x2*x3*x4*x5^5*x7 - 6*x1*x2*x3*x5^6*x7^2 - 14*x1*x2*x7^4 + 10*
    x1*x3^2*x5*x6*x7^4 + 2*x1*x3*x5^5*x7^5 + 13*x2^2*x3^4*x5^2 + 4*x2^2*x3^2*x4*x5 -
    12*x2^2*x3^2*x5^2*x7 + 10*x2^2*x4^2 + 14*x2^2*x5^2*x7^2 + 8*x2*x3^2*x5*x7^4 + 2*x2*
    x4*x7^4 + 4*x2*x5*x7^5 + 12*x7^8) : I >= 0 := by
  let terms : List (Real × Real) := [ (0, 1), (10, -x1^2*x2^2*x3/10 - x1*x2/5 + 2*x1*
    x3^2*x5*x6/5 + x1*x3*x5^5*x7/5 + x2*x3^2*x5/5 + x2*x4 + x7^4/10), (48/5, -11*x1
    ^2*x2^2*x3/48 + x1*x2 + 7*x1*x3^2*x5*x6/24 + x1*x3*x5^5*x7/4 - 7*x2*x3^2*x5/12 -
    5*x2*x5*x7/48 - 17*x7^4/24), (667/48, -7*x1^2*x2^2*x3/23 + 62*x1*x3^2*x5*x6/667 -
     132*x1*x3*x5^5*x7/667 - 316*x2*x3^2*x5/667 + x2*x5*x7 + 62*x7^4/667), (0, x2*x3*
    x5), (9289/1334, -2755*x1^2*x2^2*x3/9289 + 8622*x1*x3^2*x5*x6/9289 + 3676*x1*x3*
    x5^5*x7/9289 + 594*x2*x3^2*x5/9289 + x7^4), (57461/9289, -18159*x1^2*x2^2*x3
    /57461 - 18281*x1*x3^2*x5*x6/57461 + 4839*x1*x3*x5^5*x7/57461 + x2*x3^2*x5), (0,
    x3^4), (163073/57461, -187911*x1^2*x2^2*x3/163073 + x1*x3^2*x5*x6 - 151490*x1*x3*
    x5^5*x7/163073), (670239/163073, x1^2*x2^2*x3 - 838684*x1*x3*x5^5*x7/670239), (0,
     x1*x3*x5^5), (292711/670239, x1*x3*x5^5*x7) ]
  have : I = (terms.map (fun (p, k) => p * k^2)).sum := by
    unfold terms
    simp only [List.map_cons, List.map_nil, List.sum_cons, List.sum_nil]
    linear_combination h1
  rw [this]
  unfold terms
  simp only [List.map_cons, List.map_nil, List.sum_cons, List.sum_nil]
  positivity
```

