# OpenReview forum: "From LLM-Generated Conjectures to Lean Formalizations: Automated Polynomial Inequality Proving via Sum-of-Squares Certificates"
_ICML.cc/2026/Conference — ICML 2026 regular_

### Official Review · Reviewer_ASjC · 2026-03-03

**Soundness:** 4
**Presentation:** 3
**Significance:** 3
**Originality:** 4
**Overall Recommendation:** 5
**Confidence:** 3

**Summary:**

Current approaches—both LLM-based provers and symbolic computation baselines—struggle to scale to challenging polynomial inequality proofs. The authors propose NSPI, a neuro-symbolic pipeline that proves a polynomial is nonnegative by certifying it as a sum of squares (SOS): (1) an LLM proposes an approximate SOS conjecture close to the target polynomial, (2) a symbolic correction stage repairs this conjecture into an exact SOS certificate, and (3) a Lean proof script is generated to formally verify that the SOS expression equals the original polynomial (and hence is nonnegative). To evaluate NSPI, the authors introduce PolyIneqBench, a benchmark of 522 inequalities over 3–10 variables, including 102 problems from authoritative sources (PolyIneq-Real) and 420 synthesized instances (PolyIneq-Synth). The experiments indicate that NSPI achieves higher success rates and faster runtimes than the (wide range of) compared baselines.

**Compliance With Llm Reviewing Policy:**

Affirmed.

**Final Justification:**

I did not change my overall recommendation following the rebuttal.

My concerns were satisfactorily addressed, and no additional critical issues emerged during the rebuttal period.

**Key Questions For Authors:**

- **Table 1 (variance / stochasticity):** Since NSPI relies on an LLM, the pipeline may be non-deterministic. Could you report the variance (or confidence intervals) of success rates and runtimes over multiple runs with different random seeds / sampling settings?
- **Benchmark effects (PolyIneq-Real vs PolyIneq-Synth):** LIPS achieves higher success on PolyIneq-Real, while PolyIneq-Synth is newly introduced and fully constructed in this work. It is unclear whether NSPI’s relative advantage on the synthetic set is primarily driven by higher degrees, different coefficient distributions, or other factors. Could you include a controlled synthetic subset with lower degrees (closer to PolyIneq-Real) and compare success rate and runtime to verify that trends persist?
- **Small-$n$ regime ($n$ = 3):** For $n=3$, Z3 and Maple appear both substantially faster and more successful. What explains this gap – e.g., are these instances dominated by patterns that classical symbolic methods handle particularly well, or does NSPI incur fixed overheads (generation/correction/verification) that dominate in small settings?
- **Citations for the SDP claim:** The statement “SDP solvers typically return numerical solutions” is reasonable but should be supported with citations. Could you add appropriate references (e.g., standard SDP solver documentation or surveys noting floating-point outputs and the need for rational recovery for exact certificates)?

**Limitations:**

Limitations are not discussed in the current draft. Please add a dedicated limitations section and clarify the main constraints of your approach (e.g., assumptions, failure modes, scalability bottlenecks, and cases where NSPI is unlikely to help).

**Strengths And Weaknesses:**

**Strengths:**
- The paper effectively combines complementary strengths of LLMs, symbolic methods, and formal verification to tackle a problem that none of these components can reliably solve on its own. Each stage is substantive and involves nontrivial, original engineering and methodological choices.
- NSPI outperforms a wide range of the strongest baselines in both overall success rate and runtime.
- The paper is clearly written and well organized: it presents the high-level pipeline before diving into technical details, and appropriately relegates the most intricate material to the appendix.
- The overall NSPI paradigm appears broadly reusable: use an LLM to generate a good first approximation, apply symbolic methods to refine it into an exact solution, and then use a proof assistant to certify correctness — potentially extending beyond polynomial inequalities to other mathematical domains.

**Weaknesses:**
- The paper would benefit from a clear running example carried through the main text. This would help readers build intuition for the end-to-end pipeline while keeping the detailed derivations and technicalities in the appendix.
- While NSPI shines on harder and synthetic instances, some baselines appear stronger on simpler cases, with higher success rates on portions of PolyIneq-Real.
- The reliance on SFT and GRPO introduces a substantial training overhead: generating large-scale synthetic data and performing multi-stage fine-tuning is time- and compute-intensive.
- The paper provides little to no explanation of LIPS in the main body, even though it is the closest hybrid baseline addressing the same task; a brief description and clearer positioning would improve the comparison.

---

> ### Author Rebuttal · Authors · 2026-03-31
>
> We sincerely thank the reviewer for recognizing the complexity and substance of our modules, the strong performance in success rate and runtime, the clarity of our writing, and the reproducibility of our method. We address your concerns below:
>
> **W1: Running Example in Main Text**
>
> We agree a concrete example is vital. While Appendix H (Figures 11 & 12) provides full case studies for competition and 10-variable inequalities, we will integrate a **running example** into the revised main text to illustrate the step-by-step inputs/outputs of each module.
>
> **W2: Baseline Performance on PolyIneq-Real**
>
> The relative strength of some baselines on this subset stems from problems relying on specific "tricks" (e.g., clever substitutions) that may not align with the SOS certificate route. However, NSPI still outperforms most baselines, including SOTA general LLMs. Our goal is to provide a **scalable, end-to-end reliable pipeline** LLM Conjecture-Symbolic Correction-Lean Verification for high-dimensional problems where traditional solvers fail.
>
> **W3: SFT + GRPO Overhead**
>
> Thank you for pointing out this objective limitation. While training is the primary cost, it is a **one-time investment** that scales linearly with hardware.
> * **Data Gen:** 2h (1M samples) via multi-threading.
> * **Training:** ~10 days on 4× L40S GPUs (standard for LLM fine-tuning).
> Once trained, the model performs inference instantly without re-training.
>
> | Stage | Hardware | Time |
> | :--- | :--- | :--- |
> | Data Generation | 2× AMD EPYC 7H12 (200 threads) | ~2h |
> | SFT Training | 4× 48GB NVIDIA L40S | ~7 days |
> | GRPO Training | 4× 48GB NVIDIA L40S | ~3–4 days |
>
> **W4: Description of LIPS**
> Thank you for this helpful reminder. We will provide a more detailed description of LIPS in the related work section.
>
> **Q1: Sampling Settings (Pass@k and Temperature)**
>
> Thank you for raising this constructive concern. We have conducted a systematic evaluation from two perspectives.
>  NSPI is robust to stochasticity. Performance scales with the computational budget ($k$) and peaks at temperature $T$.
>
> | Pass@k (Real) | k=8 | k=16 | k=32 | k=64 | k=128 |
> | :--- | :---: | :---: | :---: | :---: | :---: |
> | Pass Rate | 32.27% | 40.20% | 43.14% | 44.12% | 45.10% |
>
> | Temperature | 0.3 | 0.5 | **0.7** | 0.9 |
> | :--- | :---: | :---: | :---: | :---: |
> | Pass Rate | 41.18% | 41.18% | **44.12%** | 42.16% |
>
> These two experiments show that NSPI is reasonably robust to randomness and that appropriate sampling strategies can improve performance.
>
>
> **Q2: Low-degree Synthetic Results**
>
> We additionally included a lower-dimensional synthetic subset closer to PolyIneq-Real, consisting of 60 three-variable polynomial inequalities.
> On this subset, NSPI achieves a success rate of 48.33% with an average runtime of 7.70s. This is higher than its result on the (n=4) test set (40.3%, 14.4s) and close to its performance on three-variable inequalities from PolyIneq-Real (44.10%).
> These consistent with our findings: as $n$ increases, success rates decline while runtime increases.
>
> **Q3: Performance at n=3 (Z3/Maple)**
>
> Thank you for this valuable question. We believe this phenomenon is mainly due to two factors.
> 1. Symbolic tools such as Z3 and Maple rely on Cylindrical Algebraic Decomposition (CAD), which has **double-exponential complexity**. When $n=3$, the search space remains limited, allowing efficient decomposition and solution. However, as the number of variables increases, the computation grows rapidly due to this complexity.
> 2. NSPI incurs a fixed pipeline overhead even on small-scale problems. NSPI has a constant pipeline overhead (LLM Conjecture + Symbolic Correction + Lean verification) that makes it relatively slower for trivial cases but significantly more capable as $n$ scales.
>
> **Q4: Citations for SDP Numerical Solutions**
>
> Thank you for your suggestion. We will add citations (e.g., MOSEK documentation, literature on rational recovery) to support the claim that SDP solvers typically return numerical/floating-point solutions requiring subsequent exact reconstruction.
>
> **Limitations Section**
>
> Thank you for your thoughtful suggestion. We will add a dedicated section discussing:
> 1. **Scope:** NSPI only applies to SOS-certifiable polynomials (e.g., it cannot prove the Motzkin polynomial). This is an inherent limitation of the SOS methodology, rather than a flaw specific to NSPI. In the PolyIneq-Real benchmark, 27 failed cases fall into this category.
>
> 2. **Bottlenecks:** Failures often occur at very high degrees/variables when the LLM-generated SOS conjecture contains large coefficient errors or missing/redundant terms, preventing symbolic refinement from converging to a valid exact certificate. This is the main cause of failure on difficult instances.
>
> We hope these responses clarify our work and demonstrate its substantive contributions.

---

> > ### Author Rebuttal · Reviewer_ASjC · 2026-04-03
> >
> > Thank you for the detailed response. The additional clarifications address my concerns.
> >
> > I have two remaining points:
> >
> > 1. Please incorporate the new evaluation details into the paper, especially the table reporting the overhead of SFT + GRPO.
> > 2. Following Zgof’s comment regarding support only for polynomials that admit an SOS decomposition, could you comment on how restrictive this assumption is in practice? In particular, is there any estimate or intuition for how reasonable it is to assume that a random polynomial is SOS-decomposable?

---

> > > ### Author Response · Authors · 2026-04-03
> > >
> > > Thank you very much for the follow-up and for the positive feedback. We are glad that the additional clarifications addressed your main concerns.
> > >
> > > Regarding the **first point**, we agree that these new evaluation details should be incorporated into the revised paper. In particular, we will include the table reporting the overhead of **SFT + GRPO**, so that the computational cost of the full training pipeline is presented more transparently.
> > >
> > > Regarding the **second point**, we agree that this is an important limitation to clarify. Theoretically, SOS is **sufficient but not necessary** for polynomial non-negativity, and classical results show that the two coincide only in a few special cases. Beyond these cases, SOS-certifiable polynomials form a subset of nonnegative polynomials.
> > >
> > > At the same time, to the best of our knowledge, there is **no** simple, distribution-independent estimate for how likely a “random” nonnegative polynomial is to admit an SOS decomposition, since this depends strongly on the underlying sampling model, including the degree, number of variables, coefficient distribution, and normalization. For this reason, we prefer not to make a blanket probabilistic claim.
> > >
> > > In our work, the target is not arbitrary random nonnegative polynomials, but polynomial inequalities in settings where SOS certificates are a natural proof mechanism. We will revise the paper to make this scope and its limitation more explicit.

---

### Official Review · Reviewer_Zgof · 2026-03-07

**Soundness:** 3
**Presentation:** 3
**Significance:** 2
**Originality:** 2
**Overall Recommendation:** 3
**Confidence:** 3

**Summary:**

The paper introduces NSPI, an end-to-end neuro-symbolic framework designed to automate the proving of complex, multivariate polynomial inequalities, a task that has traditionally struggled with the combinatorial explosion of symbolic methods and the data scarcity of purely neural approaches. To overcome these limitations, NSPI uses a LLM to first propose an approximate Sum-of-Squares decomposition for a given target polynomial. This heuristic conjecture is then rigorously refined through a symbolic correction module—utilizing Gauss-Newton iteration and rational recovery—to produce an exact, rational mathematical certificate. Finally, a formal verification module translates this exact SOS representation into a machine-checkable proof in Lean, guaranteeing the absolute mathematical validity of the inequality.

**Compliance With Llm Reviewing Policy:**

Affirmed.

**Final Justification:**

The detailed results provided by the authors during the rebuttal convinced me that the paper's soundness is quite good.

**Key Questions For Authors:**

1. How does the NSPI system handle, or plan to address, non-negative polynomials that fundamentally lack an SOS representation?
2. How have you verified that the model is developing generalized mathematical reasoning rather than merely learning to reverse-engineer the specific algebraic construction algorithms used in your generation scripts?
3. While the paper successfully tackles higher-dimensional problems, why does it struggle with these standard, real-world competition problems? How do you alleviate concerns that the model might be overfitting to the synthetic distributions generated for training?

**Limitations:**

Please refer to the weaknesses above

**Strengths And Weaknesses:**

# Strengths
1. Unlike previous hybrid systems that relegate LLMs to simply searching or selecting strategies, this paper elevates the LLM to function as a core symbolic conjecture engine. By having the model directly propose structural Sum-of-Squares decompositions, the authors introduce an interesting shift in neuro-symbolic design.
# Weaknesses
1. The framework is entirely predicated on the premise that an SOS decomposition is a sufficient certificate for polynomial non-negativity. However, mathematically, not all globally non-negative polynomials can be represented as a sum of squares (e.g., the Motzkin polynomial). The paper fails to address how the system would handle non-negative polynomials that fundamentally lack an SOS representation.
2. The LLM is trained on synthetic data created by generating positive semi-definite Gram matrices and subsequently deriving the polynomials. Because the model learns from the outputs of these specific algebraic construction algorithms, there is a risk that it is merely learning to reverse-engineer the generation scripts rather than developing generalized mathematical reasoning.
3. The proposed framework does not outperform the neuro-symbolic baseline LIPS system on the PolyIneq-Real dataset, which consists of 102 problems collected from international mathematics competitions and authoritative sources. Furthermore, the performance gap between NSPI and LIPS on this dataset is substantial (43.14% vs. 67.65%). While the paper aims to tackle higher-dimensional problems, this should not come at the expense of solving easier, real-world problems. The model's high performance on synthetic data, coupled with its synthetic training pipeline, amplifies concerns that it may be overfitting to the authors' specifically generated distributions.
4. The symbolic correction module relies heavily on standard mathematical optimization techniques rather than novel algorithmic contributions. Specifically, the Gauss-Newton iteration is a well-established method, yet the authors present this refinement step without citing foundational literature or explicitly acknowledging it as an integration of prior approaches.

---

> ### Author Rebuttal · Authors · 2026-03-31
>
> Thank you for your time and expertise in reviewing our submission. We appreciate your constructive suggestions and address your concerns below:
>
> ### **W1 & Q1. Incompleteness of SOS decomposition for non-negative polynomials**
>
> We entirely agree that not all non-negative polynomials admit SOS representations, and we wish to clarify that our paper **does not** claim SOS is complete for general non-negative polynomials.  The goal of NSPI is to build a more **scalable** neuro-symbolic proving framework specifically for polynomial inequalities that **possess SOS certificates**.
>
> Specifically, NSPI is a **sound but incomplete** proving framework. For instances where an SOS certificate exists, once the system successfully generates a **precise SOS representation**, the resulting proof is completely reliable. For cases that are non-negative but inherently lack a polynomial SOS representation, the current framework may fail to complete the proof. This represents the **boundary** of the certificate class and method applicability, rather than a **correctness** issue.
>
> We will clarify this more explicitly in the revised manuscript: NSPI currently focuses on **SOS-certifiable polynomial inequalities** and does not attempt to be a complete method for general non-negativity.
> Simultaneously, we agree that extending the framework to broader non-negativity certificates is an important future direction.
> Notably, while not all non-negative polynomials have a polynomial SOS representation, any non-negative real polynomial can be represented as a sum of squares of rational functions. Therefore, extending NSPI to more general certificates (e.g., rational function SOS) is a theoretically grounded follow-up research direction.
>
> ### **W2 & Q2. Generalizable structural conjecture capability**
>
> Thank you for raising this constructive concern. We provide the following clarifications.
>
> 1. **Synthetic training does not equal "memorizing scripts":** A core issue is whether the test distribution is independent of the training distribution. In fact, there are zero overlapping instances between the training set and all test benchmarks, ruling out direct memorization.
> Moreover, as shown in Appendix D.2 (Figures 7 and 8), the synthetic datasets span a wide range of algebraic complexity and structural diversity.
> 2. **Out-of-Distribution (OOD) results provide direct evidence of generalization:** As described in Section 5.3, we observed cross-group generalization during the curriculum RL phase (Figure 5). Additionally, we conducted two extra experiments:
>
> **(i) Different variable distributions:** SFT on base model using 6k 3–5 variable synthetic  data improved performance on 6–7 variables.
>
> | Model | n=6 | n=7 | n=8 |
> | :--- | :---: | :---: | :---: |
> | Base model | 11.48% | 11.67% | 16.67% |
> | After SFT | **18.03%** | **15.00%** | 16.67% |
>
> **(ii) Different data construction methods:** Training on 6k instances generated via **Computation-Driven** methods, tested on 140 **Structure-Driven** instances from PolyIneq-Synth.
>
> | Model | Data from Structure-Driven Method |
> | :--- | :---: |
> | Base model | 10.71% |
> | After SFT | **20.71%** |
>
> 3. **Architecture precludes simple reverse-engineering:** NSPI does not ask the LLM to output a final certificate. NSPI only uses the LLM to generate a floating-point SOS conjecture; the final proof is obtained through rational recovery and formal verification
>
> ### **W3 & Q3. Performance gap on PolyIneq-Real and overfitting concerns**
>
> **1. Clarification of the performance gap on PolyIneq-Real**
> We agree there is a performance gap between NSPI and LIPS on PolyIneq-Real. This is likely because many problems in PolyIneq-Real rely on ingenious constructions, variable substitutions, and classical templates that do not align with the SOS certificate route. NSPI is an **SOS-guided** framework.
> To analyze this, we tested all 102 problems in PolyIneq-Real with an SDP solver (YALMIP). **27** instances were found to lack numerical SOS certificates. Furthermore, among instances LIPS solved but NSPI did not, 14 instances also failed to find feasible SOS certificates under YALMIP.
>
> The goal of NSPI is not to be a universal prover for all inequalities or to replace methods designed for low-dimensional competition data. NSPI is a **scalable framework** for SOS-guided proving, emphasizing extensibility to higher-dimensional instances and a reliable end-to-end "Conjecture-Refinement-Verification" pipeline.
>
> **2. Clarification of overfitting concerns**
>
> While a synthetic-to-real gap is possible, overfitting is unlikely: NSPI is not a black box; the LLM only proposes an approximate conjecture. Success requires the candidate to be strictly refined and verified, which cannot be achieved simply by fitting floating-point distributions.
>
>
> We accept the limitations revealed by these results. In the final version, we will emphasize the **scope** of NSPI: it applies primarily to non-negative polynomials with SOS certificates.

---

> > ### Author Rebuttal · Reviewer_Zgof · 2026-04-03
> >
> > Thank you for providing further detailed results. I will raise my score to 3.

---

> > > ### Author Response · Authors · 2026-04-03
> > >
> > > Thank you very much for carefully reading our rebuttal and for your thoughtful feedback. We are glad that our responses have addressed the concerns. We sincerely appreciate your constructive engagement throughout this process.

---

### Official Review · Reviewer_CsCf · 2026-03-10

**Soundness:** 3
**Presentation:** 3
**Significance:** 3
**Originality:** 3
**Overall Recommendation:** 5
**Confidence:** 4

**Summary:**

The paper presents NSPI, a neuro-symbolic framework for proving polynomial inequalities. The method uses an LLM to conjecture approximate sum-of-squares decompositions, refines them symbolically into exact certificates, and then generates machine-checked Lean proofs. The paper also introduces a benchmark of 522 multivariate polynomial inequalities and reports that NSPI outperforms several symbolic, LLM-based, and hybrid baselines, especially on higher-dimensional instances.

**Compliance With Llm Reviewing Policy:**

Affirmed.

**Final Justification:**

I thank the authors for their detailed rebuttal. My primary concerns have been addressed, and I therefore raise my score to accept.

**Key Questions For Authors:**

- In their problem formulation, the authors use the vector of all monomials up to degree $d$. In practice, polynomials are often sparse, and a smaller set often suffices. In particular [2] uses a Transformer to predict smaller basis vectors in order to obtain numerical certificates for polynomials with up to 100 variables. Could NSPI be combined with their pipeline to scale to even higher dimensional cases and obtain symbolic certificates?
- In [1] the authors also introduce a dataset, consisting of 1000 polynomials. Did the authors consider including this benchmark?
- Beyond polynomial inequalities, SOS Programming has many other applications. Did the authors consider extending their pipeline to other settings, such as polynomial optimization and Lyapunov function generation in control theory?

**Limitations:**

yes

**Strengths And Weaknesses:**

Strenghts
- Using LLMs to propose SOS decompositions, while delegating exact certification to symbolic methods and Lean, is a timely and compelling neuro-symbolic design.
- The LLM does not produce the final certificate, which is instead verified symbolically and via Lean, this is elegant and rigorous
- Several sampling methods are introduced for obtaining the training data
- The method performs well empirically
- The short primer on Sum-of-Squares programming in the appendix makes the work more accessible
- The dataset PolyIneqBench is another important contribution, the field needs more benchmarks of this kind

Weaknesses
- Two relevant references on learning-augmented SOS programming are missing from the related work and should be discussed, as they appear complementary to this paper:

[1]  Li et. al, 2025. SoS1: O1 and R1-Like Reasoning LLMs are Sum-of-Square Solvers https://arxiv.org/abs/2502.20545

[2]  Pelleriti et. al, 2025: Neural Sum-of-Squares: Certifying the Nonnegativity of Polynomials with Transformers https://arxiv.org/abs/2510.13444

- While the benchmark PolyIneqBench dataset is a useful contribution, it is generated within the SOS framework itself. As a result, it is difficult to assess how well the proposed method generalizes to inequalities arising from real mathematical or application-driven settings (e.g. from control theory).
 - Beyond the number of variables, polynomial degree is a central factor in SOS complexity. Figure 7 in the appendix is helpful, but the impact of degree deserves more prominent treatment in the main paper, ideally with additional ablations.
- The paper would benefit from a more detailed analysis of failure modes. In particular, understanding when the LLM conjectures fail and when symbolic refinement breaks down would strengthen the empirical study.

Minor
- Parrilo’s PhD thesis appears to be duplicated in the bibliography.

Overall, this is a well-written paper with a compelling core idea and strong empirical results. I would be willing to increase my score if the authors adequately address the concerns above, especially those regarding related work, benchmark scope, degree-based analysis, and failure-mode analysis.

---

> ### Author Rebuttal · Authors · 2026-03-31
>
> We thank the Reviewer for the positive feedback on our framework's elegance, rigor, and the PolyIneqBench benchmark. Below are our concise responses to your concerns.
>
> **W1. Related References [1][2].**
> We thank the reviewer for pointing out these two relevant works. We will include [1] and [2] in our final Related Work.
>
> **W2. Real-World Generalization.**
> We appreciate the reviewer’s feedback regarding the benchmark’s scope. We provide the following clarification:
> 1.    **Existing Coverage of Real-World Scenarios**: PolyIneqBench already includes a real-world subset, **PolyIneq-Real**, with 102 problems drawn from international math competitions and authoritative sources, fully independent of our generation scripts. NSPI outperforms other baselines on this subset, including specialized LLM-based provers and strong general-purpose LLMs, providing evidence of generalization beyond synthetic data.
>
> 2.	**Natural Extensions to Application-Driven Fields**: Application scenarios such as Control Theory (e.g., Lyapunov function verification) are a natural extension of our work. The requirement for proving polynomial non-negativity in these fields aligns perfectly with the NSPI pipeline. We consider the application of our framework to these domains as a primary and significant direction for future work.
>
>
> **W3. Effect of Polynomial Degree ($d$).**
> We additionally evaluated NSPI and all baseline methods on a subset of PolyIneqBench grouped by degree $d$. The results are shown in the table below:
>
> | Method | d=4 | d=6 | d=8 | d=10 | d=12 | d=14 | d=16 | d=18 |
> | :--- | :---: | :---: | :---: | :---: | :---: | :---: | :---: | :---: |
> | Maple | 100% | 100% | 62.5% | 71.4% | 54.7% | 18.2% | 42.1% | 0.0% |
> | Z3 | 100% | 100% | 54.2% | 52.4% | 61.3% | 18.2% | 31.6% | 0.0% |
> | LIPS | 75.0% | 79.3% | 54.2% | 57.1% | 37.3% | 27.3% | 26.3% | 0.0% |
> | **NSPI** | 75.0% | 31.0% | **66.7%** | 52.4% | **77.3%** | 27.3% | **55.3%** | **3.1%** |
>
> Success rates generally decrease as polynomial degree increases, though not strictly monotonically, since other factors (e.g., variables and SOS terms) also affect difficulty.
>
> **W4. Failure Mode Analysis.**
> We conducted a systematic analysis of the failed cases of NSPI on PolyIneqBench and identified two main failure modes:
>
> 1. **Non-SOS structures:** The polynomial itself does not admit an SOS decomposition. In such cases, the proof may instead rely on non-SOS techniques such as AM-GM transformations, variable substitutions, or classical inequality templates. These failures reflect the scope boundary of the method rather than any soundness issue. In PolyIneq-Real, about 27 failed instances fall into this category.
>
> 2. **Conjecture quality:** The LLM-generated SOS conjecture is sometimes too inaccurate for symbolic refinement to recover an exact certificate. Typical issues include missing terms, redundant terms, or large coefficient errors. In PolyIneq-Real, about 31 failed instances fall into this category.
>
> **Q1. Incorporating Sparse Basis Prediction [2].**
> Thank you for this insightful suggestion.
> We believe the idea of [2] is naturally complementary to NSPI. A promising integration is to use the predicted sparse basis as structural guidance for the SOS conjecturing module. By restricting conjecturing to a smaller representation space, this may both improve the quality of SOS conjectures and reduce the difficulty of symbolic correction.
>
> **Q2. Evaluation on Dataset [1].**
> Direct evaluation is inappropriate because:
> 1. **Task Mismatch:** [1] evaluates binary classification accuracy; NSPI evaluates formal proof construction.
> 2. **Format Incompatibility:** [1] uses floating-point coefficients, which are incompatible with our rational reconstruction and exact refinement pipeline.
>
>
> **Extension to Lyapunov Function Synthesis.**
> For example, in control theory, the goal of Lyapunov function synthesis is to find a polynomial \\( V(x) \\) such that:
>
> (1)\\(  V(0) = 0 \\);
>
> (2)\\(V(x) > 0, \\quad \\forall x \\in \\mathcal{X} \\setminus \\{0\\} \\);
>
> (3)\\( \\mathcal{L}_f V(x) \\le 0, \\quad \\forall x \\in \\mathcal{X} \\setminus \\{0\\}\\)
>
> where the Lie derivative is defined as:
>
> \\[
> \\mathcal{L}_f V(x) = \\sum\_{i=1}^n \\frac{\\partial V}{\\partial x_i} \\cdot f_i(x)
> \\]
>
> The latter two conditions are essentially polynomial non-negativity constraints.
>
>
> These conditions are essentially polynomial non-negativity constraints. Over semi-algebraic sets, they can be reduced to SOS subproblems via Putinar’s Positivstellensatz, which fits naturally within the SOS-based architecture of NSPI.
>
> Overall, we believe the NSPI paradigm is broadly applicable to SOS-based reasoning tasks, and we will highlight these directions more clearly in the revised manuscript as important future work.
>
>
> **Minor:** We will fix the duplicate citation of Parrilo’s thesis.

---

> > ### Author Rebuttal · Reviewer_CsCf · 2026-04-01
> >
> > I thank the authors for their detailed rebuttal. My primary concerns have been addressed, and I therefore raise my score to accept.

---

> > > ### Author Response · Authors · 2026-04-01
> > >
> > > We sincerely thank you for the encouraging feedback. We are glad that our responses have addressed the concerns, and we will revise the paper accordingly.

---

### Official Review · Reviewer_AGzA · 2026-03-12

**Soundness:** 3
**Presentation:** 4
**Significance:** 3
**Originality:** 3
**Overall Recommendation:** 4
**Confidence:** 3

**Summary:**

This paper provides a neuro-symbolic polynomial inequality proving system NSPI. In particular, NSPI is based on the Sum-Of-Square (SOS) proof system, which is used to prove the non-negativity of a multivariate real polynomial. NSPI consists of three main components:

1. Neural Conjecture Module: An LLM is fine-turned (via SFT & RL) for guessing SOS decomposition for a given input polynomial

2. Symbolic Correction Module: A symbolic computation process using Newton iteration and rational recovery to get the exact SOS decomposition from those conjectured SOS decompositions generated from the first step.

3. Formal Verification Module: An automatic process that generates a Lean proof given the exact SOS decomposition.

This paper also constructs a benchmark PolyIneqBench and shows that NSPI achieves the strongest performance on this benchmark comparing with existing method.

**Compliance With Llm Reviewing Policy:**

Affirmed.

**Final Justification:**

The rebuttal addressed my questions and somewhat reinforced my previous assessment. Thus, I raised my confidence by 1 (2->3).

**Key Questions For Authors:**

Did the LLM in NSPI use CoT? If not, why would you expect a LLM without CoT might be effective at generating SoS decomposition?

**Limitations:**

I hope this paper could explain what the major performance bottleneck of this NSPI system is, since its performance is still far from perfect on the benchmark. What can be improved in the future?

**Strengths And Weaknesses:**

Strength:

1. Each component of this NSPI is technically solid and it scales better than existing method when the number of variables increases.

2. This paper is very well-written.

Weakness: If I understand correctly, the LLM here is not a reasoning model using CoT. I'm kind of doubtful that even after fine-turning, a LLM without CoT could be a good at guessing SoS decomposition.

---

> ### Author Rebuttal · Authors · 2026-03-31
>
> We sincerely thank you for your thoughtful review, as well as for recognizing the reliability and superior performance of each module and your affirmation that our paper is well-written. We also appreciate your constructive suggestions and have addressed your concerns below.
>
>
> ### **Q1. Whether CoT was used**
>
> We appreciate this valuable question. Our method does not rely on explicit natural-language Chain-of-Thought (CoT). Our focus is on structural conjecture generation rather than having the model present a step-by-step proof process in natural language. Experimental results demonstrate that even without explicit CoT, the model can still learn conjectures that are helpful for subsequent refinement. We agree that exploring whether CoT or other stronger reasoning scaffolds can further improve conjecture quality is a highly valuable direction for future work.
>
>
>
> ### **W1 & Q1. Why it can be effective without CoT**
>
> Thank you for raising this important point. We would like to explain why this design is reasonable:
>
> 1.  **The nature of the task determines that CoT is not a prerequisite:** The core value of CoT lies in tasks requiring multi-step symbolic reasoning (such as mathematical proofs or logical inference). However, the role of the LLM in NSPI is fundamentally different: the LLM acts as a **structural conjecture generator** that proposes an approximate SOS decomposition for a given polynomial. This task is not entirely identical to open-ended natural language multi-step reasoning. More accurately, it is a mapping from polynomial structures to SOS decomposition structures; therefore, it does not strictly require explicit CoT to be effective. The explicit reasoning chains that CoT excels at do not constitute the core requirement here.
> 2.  **Symbolic correction provides fault tolerance for LLM output quality:** The design philosophy of NSPI is that the LLM does not need to directly output a finalized, correct formal proof. It only needs to provide a candidate that is helpful for subsequent modules; thereafter, this candidate must undergo symbolic correction and Lean verification. Thus, even without explicit CoT, the model is effective within the system as long as it can generate a structural approximation good enough for subsequent modules to refine into an exact certificate.
>
>
>
> ### **Regarding performance bottlenecks and future improvements**
>
> Thank you for this constructive suggestion. Regarding why the system has not yet reached perfect performance on current benchmarks, we agree that this could be articulated more clearly in the paper. The main bottlenecks of the current system lie in the following two areas:
>
> 1.  **Initial conjecture quality of the neural module limits overall success rate:** The overall success rate of the system is still largely constrained by the quality of the initial conjecture proposed by the neural module. When the target polynomial has a high degree or many variables, the approximate SOS conjectures output by the LLM often suffer from large coefficient errors or missing monomials, preventing subsequent symbolic refinement from obtaining a valid rational certificate. This is the primary cause of failure on high-difficulty instances. To address this, future work could introduce richer training samples of high-degree and high-dimensional polynomials to enhance the LLM's structural estimation capabilities, or explore stronger reasoning scaffolds (such as CoT) to guide the model in explicitly analyzing coefficient structures before outputting a conjecture.
> 2.  **Cases where the target polynomial lacks an SOS decomposition:** For certain polynomials, the natural proof path for non-negativity relies on non-SOS techniques such as AM-GM transformations, change of variables, or Positivstellensatz representations. The current conjecture format, centered on SOS decomposition, cannot fully capture these structures. Future work will explore introducing alternative paths such as Positivstellensatz representations into the pipeline to expand the system’s applicability. Furthermore, while not all non-negative polynomials possess a polynomial SOS representation, any non-negative real-coefficient polynomial can be represented as a sum of squares of rational functions. Therefore, extending NSPI to rational function SOS representations is also a theoretically promising future direction.
>
>
> Once again, we thank you for your valuable feedback and suggestions. We hope these responses address your concerns and clarify any doubts.

---

> > ### Author Rebuttal · Reviewer_AGzA · 2026-04-03
> >
> > I want to thank the authors for the detailed response. I agree that CoT may not be the right way, and the authors also confirmed that the LLM conjecture module is one of the bottlenecks of the performance, as I expected. Overall, I still feel the current LLM conjecture module is the weak point in the design of NSPI. I won't update the score, but I will raise my confidence by 1.

---

> > > ### Author Response · Authors · 2026-04-04
> > >
> > > Thank you very much for your careful reading of our rebuttal and for your thoughtful feedback. We are pleased that our clarifications were helpful in addressing your concerns, and we sincerely appreciate your time and constructive engagement throughout the review process.

---

### Decision · Program_Chairs · 2026-04-30

**Decision:**

Accept (regular)

**Comment:**

This paper proposes NSPI, a neuro-symbolic framework that combines LLM-based structural conjecture generation with symbolic SOS certificate refinement and end-to-end Lean formal verification for automated polynomial inequality proving. Reviewers agreed on the elegance of using an LLM not to produce the final proof but as a structural approximation engine, with exact certification delegated to symbolic methods and Lean. The empirical results are also sufficient, achieving state-of-the-art performance on a newly proposed benchmark. The main concern is from Reviewer Zgof (initial Weak Reject), who raised the most substantive concerns around SOS incompleteness, synthetic overfitting, and the PolyIneq-Real performance gap; after the rebuttal, the reviewer acknowledged concerns as fully resolved. The paper is overall technically solid and well-written, and it made a solid contribution to this field. The authors should discuss the limitations/weaknesses pointed out by reviewer Zgof and add the additional experimental results into the final revision.